# Vpr counteracts the restriction of LAPTM5 to promote HIV-1 infection in macrophages

Li Zhao[1,2], Shumei Wang[1,2], Meng Xu[3], Yang He[3], Xiaowei Zhang[1,2], Ying Xiong[1,2], Hong Sun[1,2], Haibo Ding[1,2], Wenqing Geng[1,2,4], Hong Shang[1,2,4,5] & Guoxin Liang ⬤ [1,2,3,4 ✉]

The HIV-1 accessory proteins Vif, Vpu, and Nef can promote infection by overcoming the inhibitory effects of the host cell restriction factors APOBEC3G, Tetherin, and SERINC5, respectively. However, how the HIV-1 accessory protein Vpr enhances infection in macrophages but not in CD4[+] T cells remains elusive. Here, we report that Vpr counteracts lysosomal-associated transmembrane protein 5 (LAPTM5), a potent inhibitor of HIV-1 particle infectivity, to enhance HIV-1 infection in macrophages. LAPTM5 transports HIV-1 envelope glycoproteins to lysosomes for degradation, thereby inhibiting virion infectivity. Vpr counteracts the restrictive effects of LAPTM5 by triggering its degradation via DCAF1. In the absence of Vpr, the silencing of LAPTM5 precisely phenocopied the effect of Vpr on HIV-1 infection. In contrast, Vpr did not enhance HIV-1 infection in the absence of LAPTM5. Moreover, LAPTM5 was highly expressed in macrophages but not in CD4[+] T lymphocytes. Re-expressing LAPTM5 reconstituted the Vpr-dependent promotion of HIV-1 infection in primary CD4[+] T cells, as observed in macrophages. Herein, we demonstrate the molecular mechanism used by Vpr to overcome LAPTM5 restriction in macrophages, providing a potential strategy for anti-HIV/AIDS therapeutics.

[1] Department of Laboratory Medicine, Key Laboratory of AIDS Immunology of Ministry of Health, The First Affiliated Hospital, China Medical University, Shenyang, China. [2] National Clinical Research Center for Laboratory Medicine, The First Affiliated Hospital of China Medical University, Shenyang, China. [3] Research Institute for Cancer Therapy, The First Affiliated Hospital, China Medical University, Shenyang, China. [4] Key Laboratory of AIDS Immunology, Chinese Academy of Medical Sciences, Shenyang, China. [5] Collaborative Innovation Center for Diagnosis and Treatment of Infectious Diseases, Hangzhou, China. ✉email: gxliang@cmu.edu.cn

HIV-1 can infect several immune cell types but primarily targets CD4[+] T lymphocytes and macrophages. Unlike the Gag, Pol, Rev, and Env proteins, essential for HIV-1 replication, Vpr, Vif, Vpu, and Nef are known as accessory proteins because they are not essential for HIV replication in certain cell types[1]. However, these proteins are necessary for infection in vivo as well as for efficient viral replication in different immune cells in vitro[2]. Vif, Vpu, and Nef counteract the host restriction factors APOBEC3G, Tetherin, and SERINC5 to promote HIV-1 infection, respectively[1,3–6]. Vpr, a small basic 14-kDa protein of 96 amino acids, is a nucleocytoplasmic shuttling protein[7,8] required for virus replication in vivo[9,10]. It is conserved across human (HIV-1 and HIV-2) and primate lentiviruses and specifically incorporated into viral particles by interactions with the p6 domain of Gag. Vpr-deficient HIV-1 showed a significant defect in replication in primary macrophages but not in established cell lines, such as Jurkat, HeLa, and 293T or Vpr's effect is not robust in primary CD4[+] T cells[11,12]. However, although various functions of Vpr have been reported, such as HIV-1 long terminal repeat promoter transactivation, nuclear import of the pre-integration complex, cellular apoptosis, cell cycle arrest, and activation of the DNA damage response[13–15], the role of Vpr in promoting HIV-1 infection in macrophages remains unknown[16]. Similar to other accessory proteins, Vpr is presumed to function by recruiting cellular factors to a cellular E3 ligase complex containing DDB1-CUL4-associated factor 1 (DCAF1)[17,18]. Although several host factors depleted by Vpr have been identified, their connection to Vpr-associated cell biological phenotypes and role in regulating viral replication in vivo remains unclear[15,19–25]. Furthermore, recent findings derived from replication-competent HIV-1 infection of primary human monocyte-derived macrophages (MDMs) suggest that Vpr plays no role in the first round of infection instead affects the production of the viral envelope glycoprotein Env by overcoming unknown macrophage-specific restriction factors in a DCAF1-dependent manner[26]. Nevertheless, the molecular mechanisms underlying the function of Vpr in HIV-1 infectivity still remain to be elucidated.

Here, we show that LAPTM5 protein inhibits HIV-1 progeny infectivity by transporting viral Env to the lysosome for degradation in macrophages. However, Vpr could induce polyubiquitination of LAPTM5 protein to prompt its degradation, consequently enhancing HIV-1 infection. The Vpr-induced degradation of LAPTM5 relies on DCAF1 and the proteasome. Depletion of DCAF1 or inhibition of proteasome compromises the Vpr-induced LAPTM5 degradation. Moreover, ectopic expression of LAPTM5 could significantly inhibit HIV-2 and SIV, but not feline immunodeficiency virus (FIV) or murine leukemia virus (MLV) infectivity, suggesting LAPTM5 to be a potential anti-HIV agent.

## Results

**Vpr overcomes LAPTM5 to enhance HIV-1 infection.** We first investigated the role of Vpr in the virion infectivity of macrophages by comparing the viral production of wild-type HIV-1 (BaL, AD8, 89.6) with its Vpr-defective counterpart. In agreement with previous findings[26], we observed Vpr to have no apparent effect on viral production in the first round of infection of MDMs in our experimental settings (Supplementary Fig. 1a). In fact, Vpr can significantly enhance HIV-1-spreading infection in MDMs[26]. Following this, we transduced primary MDMs in vitro derived from independent healthy donors with a lentiviral FLAG-tagged Vpr expression vector to explore the mechanisms adopted by Vpr to counteract the potential specific restriction factor(s) in macrophages. We posited that Vpr would trigger the proteasome-mediated degradation of host restriction factor(s); therefore, we treated MDMs with MG132 and performed immunoprecipitation (IP) assays to isolate Vpr-interacting proteins (Supplementary Fig. 1b). We set the low threshold at 30 peptides and compared the candidate proteins with or without MG132 treatment. LAPTM5[27–29] emerged as a potential Vpr-interacting factor because it precipitated with Vpr in the presence of MG132 but failed to precipitate in the absence of MG132 (Supplementary Table 1), suggesting LAPTM5 as a potential factor degraded by Vpr. To validate whether Vpr induces LAPTM5 degradation, we infected MDMs with wild-type or Vpr-defective HIV-1$_{AD8}$ in the presence or absence of Efavirenz (EFV) or raltegravir, which block HIV-1 productive infection. As a result, we found that Vpr could induce degradation of the LAPTM5 protein in primary MDMs (Supplementary Fig. 1c), despite the fact that its transcript levels were unaffected (Supplementary Fig. 1d). Furthermore, this degradation must rely on HIV-1 productive infection, as it was completely suppressed by EFV or raltegravir treatment.

Because our data suggest that Vpr counteracts LAPTM5 by inducing its degradation, we decided to investigate whether the Vpr-induced degradation of LAPTM5 promotes the spread of HIV-1 infection among primary MDMs. Therefore, we aimed to distinguish the effects of Vpr on different HIV-1 replication stages in the presence or absence of LAPTM5 (Supplementary Fig. 1e). From the results, we noted obvious differences in virion production between normal LAPTM5-positive MDMs infected with the same viral stocks of either wild-type or Vpr-defective HIV-1$_{AD8}$, particularly at low inoculum levels (Fig. 1a). The impact of Vpr on virus production was only obvious in the presence of LAPTM5. In the absence of LAPTM5, we observed almost no effect of Vpr on virion production in MDMs, in which wild-type HIV-1 or its Vpr-defective counterpart were allowed to spread to saturation over 18 days. To confirm these results, we infected LAPTM5-depleted or normal MDMs (Supplementary Fig. 1f) with other wild-type or Vpr-defective HIV-1 isolates (BaL and 89.6). Vpr consistently promoted HIV-1 infection in the presence, but not absence, of LAPTM5 during 18 days of infection spreading (Fig. 1b, c). The data obtained from the experiments were derived from a single measurement and repeated in three different healthy donors, suggesting that LAPTM5 may inhibit HIV-1 infection and Vpr can compromise the inhibition. Crucially, when we used a lentiviral shRNA targeting 3'-UTR of the LAPTM5 transcript, we found that the requirement for Vpr returned in shRNA-mediated LAPTM5-depleted MDMs, which were transduced with an LAPTM5-expression lentiviral vector during the spread of HIV-1 infection (Fig. 1d, e). The exogenous LAPTM5 expression was not affected by this shRNA because it did not affect LAPTM5 expression derived from its expression vector, into which the LAPTM5 open reading frame was cloned. Taken together, LAPTM5 has a negative effect on the spread of HIV-1 infection of primary MDMs, whereas Vpr can overcome the inhibitory effect of LAPTM5.

**Expression of LAPTM5 in MDMs.** The LAPTM5 family is known to have three members: LAPTM5, LAPTM4A, and LAPTM4B. In our experiments, we found that LAPTM5, but not LAPTM4A or 4B, precipitated with Vpr in primary MDMs. Therefore, we determined the expression profile of LAPTM5 protein family in various host cells. Of the three proteins, only LAPTM5 was abundantly expressed in primary monocytes, MDMs, and monocyte-derived dendritic cells (MDDCs). However, LAPTM5 expression was not detected in established cell lines, such as 293T, HeLa, and Jurkat cells (Supplementary Fig. 2a, b). Similarly, LAPTM5 transcripts was found to be nearly

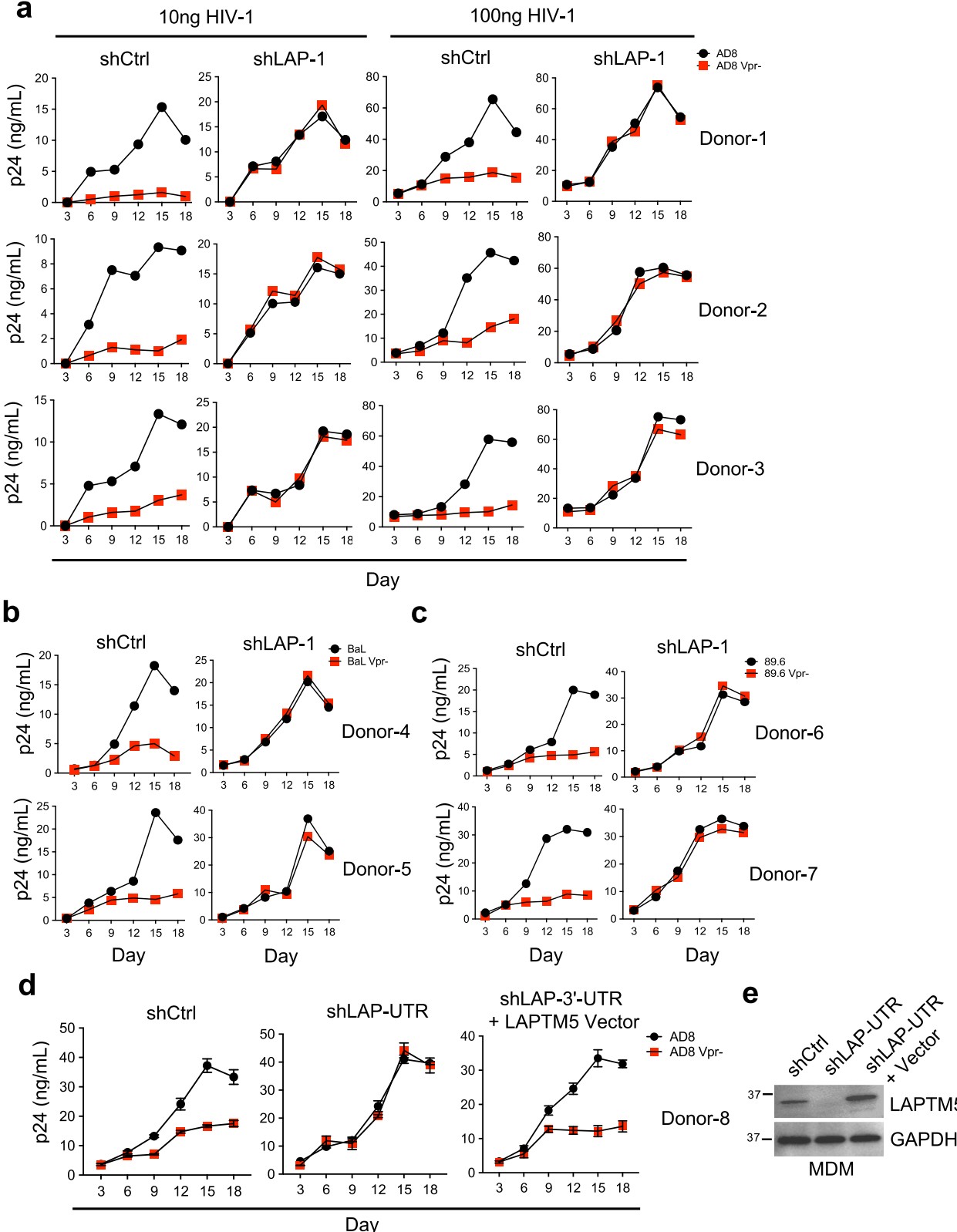

100-fold lower in stimulated CD4+ T cells than in primary MDMs or MDDCs. Moreover, only LAPTM5 was induced and highly expressed in the PMA-stimulated THP-1 cells (Supplementary Fig. 2c, d). These results suggest that LAPTM5 is the only protein in the LAPTM5 family that is a macrophage-specific restriction factor.

**LAPTM5 restricts HIV-1 infectivity**. We explored the mechanisms by which LAPTM5 restricts HIV-1 infection. At first, we observed that LAPTM5 overexpression had no effect on Vpr-defective HIV-1 during the early phase of viral replication (Fig. 2a), suggesting that inhibition by LAPTM5 occurs in later phases. To functionally study the effect of LAPTM5 on viral

**Fig. 1 Vpr counteracts LAPTM5 to promote HIV-1 infection in macrophages. a** Lentiviral shRNA-transduced MDMs were infected with 10 or 100 ng of wild-type or Vpr-defective HIV-1$_{AD8}$ for 18 days. Before infection, the aliquoted MDMs were lysed for western blotting to assess LAPTM5 and GAPDH expression (Supplementary Fig. 1e). Viral production was measured by p24 ELISA at the indicated time points, and the data were derived from single measurement. **b**, **c** Lentiviral shRNA-transduced MDMs were infected with 100 ng of wild-type or Vpr-defective HIV-1 for 18 days. Before infection, aliquoted cells were lysed for western blotting to assess LAPTM5 and GAPDH expression (Supplementary Fig. 1f). Viral production was measured using p24 ELISA at the indicated time points, and the data were derived from single measurement. **d**, **e** MDMs were transduced with lentiviral shRNA targeting the 3'-UTR of *LAPTM5* transcripts or control, and the shLAPTM5-depleted MDMs were further transduced with or without a lentiviral expression vector of non-tagged LAPTM5. MDMs were further infected with 100 ng of wild-type or Vpr-defective HIV-1$_{AD8}$ for 18 days and viral production was measured by p24 ELISA at the indicated time points. Data presented are means ± SD of three independent measurements (**d**). Before HIV-1 infection, the aliquoted shRNA-transduced MDMs were lysed for western blotting to assess LAPTM5 and GAPDH expression (**e**). All western blot data are representative of three independent experiments; their full-size images are presented in the Source Data.

infection in the later phases, we overexpressed LAPTM5 in HeLa cells (endogenous LAPTM5 protein was not detected), and examined the resulting HIV-1 infectivity. LAPTM5 resulted in a dramatic reduction in Vpr-defective HIV-1 infectivity, when combined with the expression of various HIV-1 CXCR4-, CCR5-, or dual-tropic Env (Fig. 2b). In contrast, no inhibitory effect of LAPTM5 on VSV-G-pseudotyped HIV-1 was observed. To explore whether Vpr counteracts the inhibitory effect of LAPTM5 on HIV-1 infectivity, we overexpressed Vpr along with LAPTM5 and determined HIV-1 infectivity in producer cells. Vpr significantly diminished the inhibitory effect of LAPTM5 on Vpr-defective CXCR4-tropic NL4-3 and CCR5-trophic BaL (Fig. 2c, d). In addition, LAPTM5, but not the monitoring GFP protein, was degraded in the presence of Vpr, indicating that Vpr induced the degradation of LAPTM5. Furthermore, we titrated LAPTM5 expression to examine the infectivities of wild-type and Vpr-defective HIV-1. Consequently, at the low concentration of LAPTM5 expression vector, LAPTM5 was able to inhibit Vpr-defective HIV-1 infectivity and had no significant effect on wild-type HIV-1 (Fig. 3a). Furthermore, the wild-type and Vpr-defective HIV-1 were similarly and dramatically inhibited by the high dose of LAPTM5, indicating that the endogenous Vpr in HIV-1 was unable to counter such high levels of LAPTM5.

**Vpr induces LAPTM5 degradation via DCAF1.** To follow, we studied the mechanism Vpr adopts to promote the degradation of LAPTM5. We observed that Vpr dramatically decreased LAPTM5, but not GFP protein expression, in the absence of HIV-1 (Supplementary Fig. 3a). We then overexpressed LAPTM5 in an IRES-GFP expression vector and examined the effects of Vpr from different HIV-1 isolates (BaL, 89.6, AD8, NL4-3, and Yu2) on the reduction of LAPTM5 protein. We found that Vpr clearly decreased levels of LAPTM5 protein (Fig. 3b); however, the GFP protein, which is bicistronically expressed with LAPTM5, was unaffected, thereby confirming that the decrease in LAPTM5 is due to the degradation of the protein. As Vpr is conserved across HIV-1 and primate lentivirus SIV, we overexpressed Vpr from SIV$_{mac239}$ and HIV-2$_{Rod}$; however, there was no obvious decrease in LAPTM5 protein levels (Supplementary Fig. 3b). Notably, Vif from HIV-1 and SIV can efficiently degrade the APOBEC3G enzymes of their host species but are unable to counteract homologous APOBEC3G from other species[30–32]. It is possible that Vpr is similar to Vif in this regard. To test this assumption, we examined Vpr derived from SIV$_{mac239}$ and SIV$_{agm}$ along with rhesus macaque LAPTM5 and found that both Vpr homologs can efficiently promote the reduction of rhesus macaque but not human LAPTM5 protein (Supplementary Fig. 3c).

Given the fact that Vpr-induced degradation of its targets relies on proteasomes[21,22,24], we hypothesized that Vpr also requires proteasomes to degrade LAPTM5. To assess this, we examined the Vpr-triggered degradation of LAPTM5 in the presence or absence of MG132 and found that MG132 treatment completely

abolished the degradation of LAPTM5 (Fig. 3c). Therefore, we concluded that Vpr induces LAPTM5 degradation in a proteasome-dependent manner.

Interestingly, a mutant Vpr (Vpr$^{Q65R}$), defective in promoting HIV-1 infectivity because it does not interact with DCAF1[26], was unable to induce LAPTM5 degradation (Supplementary Fig. 4a, b), although we showed that Vpr$^{Q65R}$ colocalized and interacted with LAPTM5 proteins in the cytoplasm of HeLa cells (Supplementary Fig. 4c, d). Notably, Vpr protein has been shown to function as a nucleocytoplasmic shuttling protein[7]. In contrast with HeLa cells, we observed that Vpr could also be localized in the cytoplasm of primary MDMs (Supplementary Fig. 4e). Moreover, we observed that Vpr could be colocalized with LAPTM5 in the cytoplasm of primary MDMs (Supplementary Fig. 4f), suggesting that Vpr counteracts LAPTM5 in the cytoplasm. When we explored whether wild-type Vpr and its mutant Vpr$^{Q65R}$ could induce polyubiquitination of LAPTM5 to prompt its degradation, we found that only wild-type Vpr promoted polyubiquitination of LAPTM5 (Supplementary Fig. 5a, b), suggesting that Vpr, but not the Vpr$^{Q65R}$ mutant, can enhance ubiquitination of LAPTM5 to induce its degradation. On the contrary, Vpr$^{Q65R}$ can bind to LAPTM5, but it cannot promote LAPTM5 polyubiquitination to induce degradation. Similarly, we observed that neither SIV nor HIV-2 Vpr was able to induce polyubiquitination of human LAPTM5 (Supplementary Fig. 3d), which may explain why neither of them can promote the degradation of human LAPTM5. Notably, Vpr has been suggested to bind to DCAF1[26], and we observed that precipitating LAPTM5 precipitated both Vpr and its cofactor, DCAF1 (Supplementary Fig. 5c); this interaction between LAPTM5 and DCAF1 depended on the presence of Vpr, suggesting that Vpr triggers LAPTM5 degradation in a DCAF1-dependent manner. To clarify this, we employed RNAi to silence DCAF1 in HeLa cells and primary MDMs, and we found that DCAF1 silencing dramatically compromised the Vpr-induced LAPTM5 degradation (Fig. 3d, e), as well as polyubiquitination of LAPTM5 (Supplementary Fig. 5d). Collectively, these results indicate that Vpr recruits DCAF1 to promote the polyubiquitination of LAPTM5, resulting in its proteasome-mediated degradation.

**LAPTM5 family members restrict HIV-1, HIV-2, and SIV infection.** We mechanistically probed how LAPTM5 combats HIV-1 infection by investigating LAPTM5 and the other LAPTM family members LAPTM4A and 4B. The overexpression of LAPTM5 resulted in a 257- to 664-fold reduction in infection by CXCR4-, CCR5-, or dual-tropic Vpr-defective HIV-1, whereas the production of HIV-1 capsid was intact (Fig. 4a–c). Similarly, LAPTM4A overexpression significantly restricted HIV-1 infectivity. However, LAPTM4B showed only a mild restrictive effect as compared with LAPTM5 and LAPTM4A. Regardless of the presence of Vpr, LAPTM5 overexpression also showed a robust inhibitory effect on wild-type HIV-1 infectivity (Supplementary Fig. 6a), suggesting that the effect of Vpr expressed from HIV-1 was masked

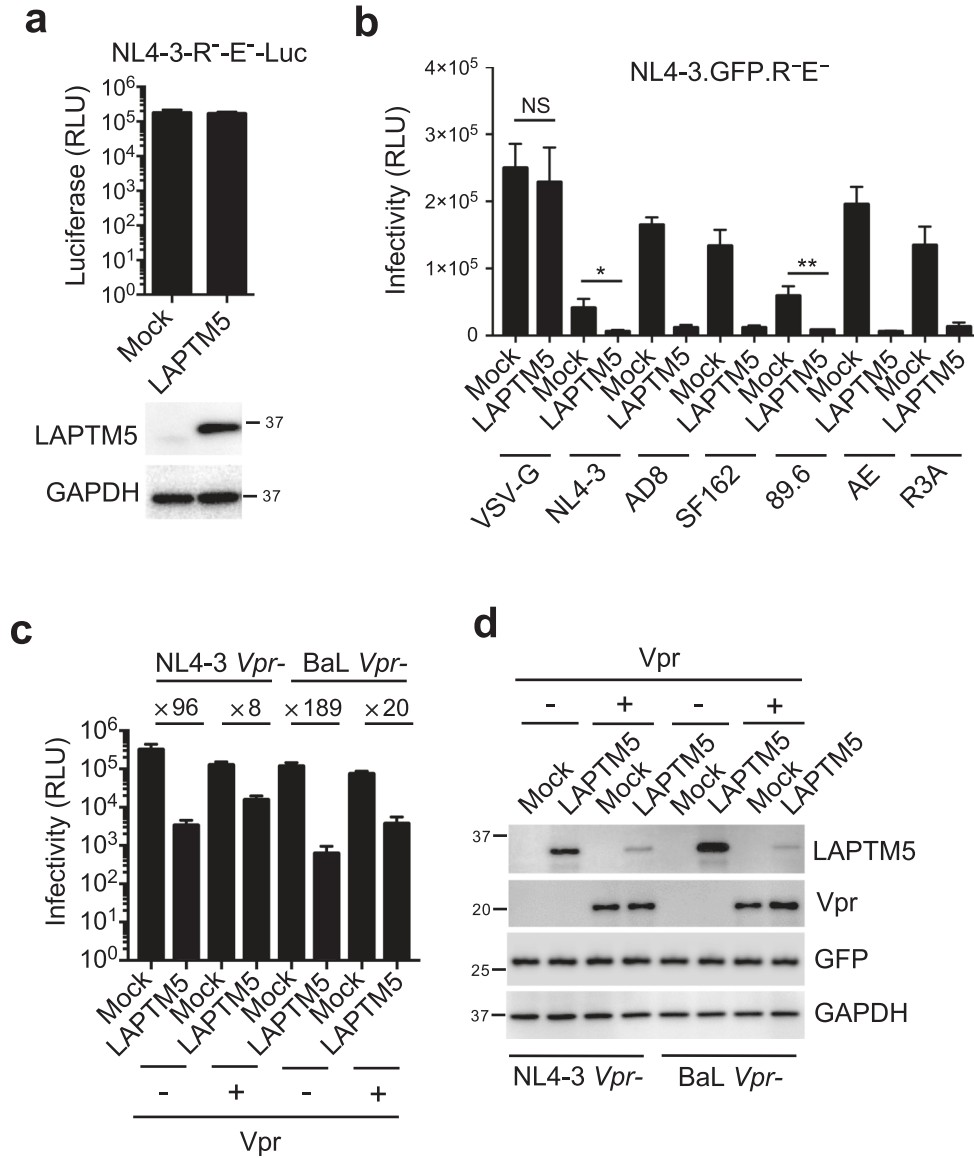

**Fig. 2 LAPTM5 restricts HIV-1 infectivity. a** HeLa cells were transfected with FLAG-tagged LAPTM5 or mock expression constructs. At 24 h after transfection, cells were infected with 100 ng of Vpr-defective HIV-1$_{NL4-3.Luc.R-E-}$ (VSV-G). At 2 dpi, cells were lysed to measure luciferase reporter activity, and western blotting was performed to assess exogenous protein expression. Data are plotted as mean ± SEM of three independent experiments. **b** HeLa cells were cotransfected with FLAG-tagged LAPTM5 or mock expression constructs along with a Vpr-defective pNL4-3.GFP.R⁻E⁻ reporter vector combined with various expression constructs of HIV-1 ×4-, R5-trophic, or dual-trophic Env or VSV-G as indicated. Two days after transfection, TZM-bl indicator cells were used to measure HIV-1 infectivity. *$P < 0.05$, **$P < 0.01$; NS, not significant (two-tailed, unpaired Student's $t$-test), data are plotted as mean ± SEM of three independent experiments. **c, d** HeLa cells were cotransfected with non-tagged LAPTM5 or mock expression constructs along with Vpr-defective HIV-1 proviral vectors (NL4-3 Vpr- or BaL Vpr-) and a monitoring expression plasmid of GFP in the presence or absence of a FLAG-tagged Vpr expression vector derived from NL4-3. Two days after transfection, TZM-bl indicator cells were used to measure HIV-1 infectivity (**c**). Cells were lysed for western blotting to assess LAPTM5, Vpr, GFP, and GAPDH expression (**d**). Data are plotted as mean ± SEM of three independent experiments. All western blot data are representative of three independent experiments; their full-size images are presented in the Source Data.

by the high levels of LAPTM5 protein. In contrast to LAPTM5, Vpr promote the degradation of LAPTM4A and 4B proteins, indicating that LAPTM5 is the only member in this family that Vpr can target.

Furthermore, we examined the effects of overexpressing LAPTM5, 4A, and 4B on various wild-type replication-competent primate lentiviruses (HIV-2$_{Rod}$, SIV$_{mac239}$, and SIV$_{agm}$) and non-primate viruses (FIV and gamma-retrovirus MLV). LAPTM5 was found to significantly inhibit HIV-2$_{Rod}$, SIV$_{mac239}$, and SIV$_{agm}$ virion infectivity (Supplementary Fig. 6b–d); however, none of them had any effect on the infectivity of FIV and MLV (Supplementary Fig. 6e, f), implying that LAPTM5 might target

primate lentiviruses. In addition, the overexpression of rhesus macaque or chimpanzee LAPTM5 could inhibit the infectivities of replication-competent SIV$_{mac239}$ and SIV$_{agm}$ (Supplementary Fig. 6g, h), indicative of a conserved antiviral activity for LAPTM5 across species. Therefore, LAPTM5 might provide widely antiviral activities in innate immunity.

**Lysosomal degradation of Env by LAPTM5.** We investigated which step of the HIV-1 life-cycle is blocked by LAPTM5. Because HIV-1 (NL4-3, BaL, and AD8) virion capsid production in cell culture was not affected by LAPTM5, we examined the

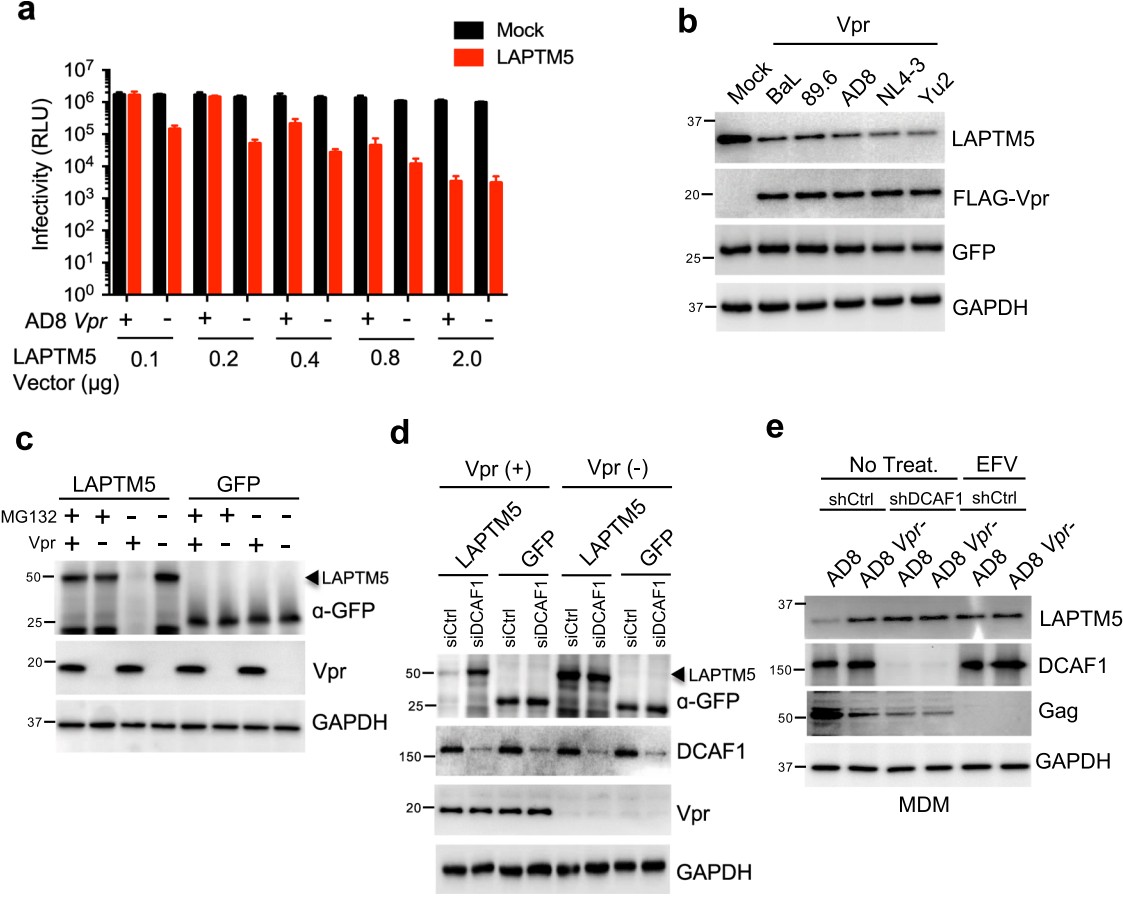

**Fig. 3 LAPTM5 restricts HIV-1 infectivity and Vpr counteracts the restriction. a** HeLa cells were cotransfected with FLAG-tagged LAPTM5 or mock expression constructs at the indicated doses along with wild-type or Vpr-defective HIV-1_AD8 proviral vectors. Two days after transfection, TZM-bl indicator cells were used to measure HIV-1 infectivity. Data are plotted as mean ± SEM of three independent experiments. **b** HeLa cells were cotransfected with a bicistronic construct containing an IRES to coexpress LAPTM5 and GFP (LAPTM5-IRES-GFP) with or without FLAG-tagged Vpr expression vectors derived from different HIV-1 isolates. Two days after transfection, cells were lysed to assess LAPTM5, Vpr, GFP, and GAPDH expression by western blotting. **c** HeLa cells were cotransfected with GFP-tagged LAPTM5 or mock (GFP) expression constructs with or without a FLAG-tagged Vpr expression vector in the presence or absence of MG132 (1.5 μM treatment for the last 12 h). Two days after transfection, cells were lysed to assess LAPTM5-GFP, GFP, Vpr, and GAPDH expression by western blotting. **d** HeLa cells were cotransfected with GFP-tagged LAPTM5 or mock (GFP) expression constructs with or without a FLAG-tagged Vpr expression vector in the presence of siRNA against DCAF1 or control siRNA. Two days after transfection, cells were lysed to assess LAPTM5-GFP, GFP, DCAF1, Vpr, and GAPDH expression by western blotting. **e** Lentiviral shRNA-transduced MDMs were infected with 500 ng wild-type or Vpr-defective HIV-1_AD8 in the presence or absence of EFV (0.3 μM). At 6 dpi, cells were lysed to assess LAPTM5, DCAF1, Gag, and GAPDH expression by western blotting. All western blot data are representative of three independent experiments; their full-size images are presented in the Source Data.

levels of virion-produced Env in culture. The levels of Env gp120, but not Gag p24, of HIV-1 virions dramatically reduced ranging from 43- to 146-fold reduction (Fig. 5a). Accordingly, HIV-1 Env in the cytoplasm was also reduced, whereas Gag was intact, in the presence of LAPTM5 (Fig. 5b), demonstrating that LAPTM5 inhibits HIV-1 infectivity by reducing Env. Therefore, we assumed that LAPTM5 might function as a carrier, transporting HIV-1 Env to the lysosome for degradation. To test this assumption, we first analyzed the subcellular localization of LAPTM5 and observed that LAPTM5 was colocalized with the lysosomal marker LAMP1 protein in HeLa, THP-1 cells (Supplementary Fig. 7a, b), and primary MDMs (Supplementary Fig. 7c), indicating that LAPTM5 might transfer HIV-1 Env to lysosomes. To confirm whether LAPTM5 transfer Env to lysosome for degradation, we next used lysosomal inhibitors bafilomycin A1 (BFLA), chloroquine (CLQ), or NH4Cl to determine the anti-HIV activity of LAPTM5. None of the inhibitors abolished the inhibitory effect of LAPTM5 on HIV-1 infectivity (Supplementary Fig. 8). Despite LAPTM5 expression, Env levels in cell lysates were restored in the presence of lyso-

inhibitors, whereas the Gag protein was unaffected (Fig. 5c). However, in culture, the virion gp120 levels were not restored in the presence of lysosomal inhibitors (Fig. 5d). Therefore, we assumed that HIV-1 Env was still trapped in the lysosome when treated with lysosomal inhibitors. Therefore, Env could not be released via the plasma membrane for progeny assembly and consequently virion glycoprotein levels were not restored. Lysosomal inhibition may not affect the ability of LAPTM5 to transport viral Env to the lysosome. To confirm this, we used immunofluorescence assays and found that the rescued Env colocalized with LAPTM5 in the presence of lysosomal inhibition, whereas Gag proteins were not affected (Fig. 5e). In contrast, Env and Gag proteins were not affected in the absence of LAPTM5, regardless of the presence of lysosomal inhibitors (Fig. 5f). To test whether the restored Env is localized in lysosome, we enriched lysosomes and found that the inhibitor treatment could upregulate Env levels in the lysosome only when LAPTM5 was present (Fig. 5g). Similarly, once the lysosomal inhibitor was absent, the LAPTM5-transferred Env was not detected in the lysosomes due

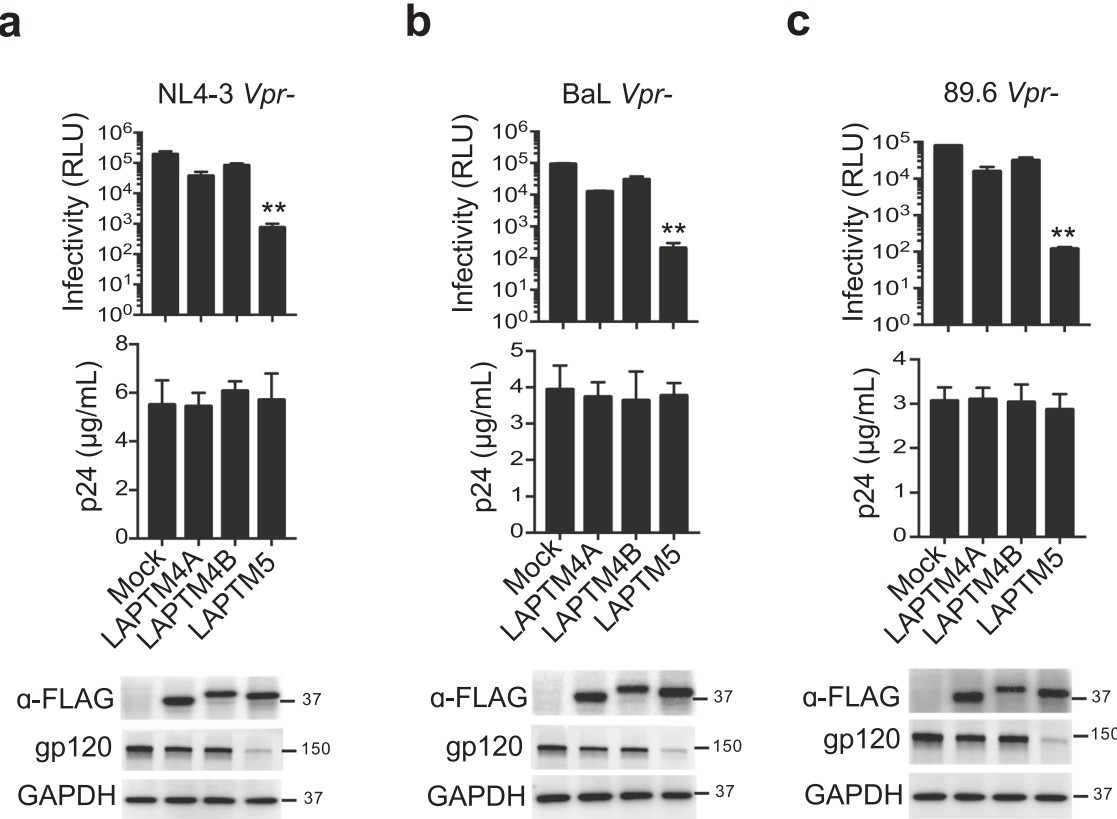

**Fig. 4 LAPTM5 and LAPTM4A restrict HIV-1 infectivity. a–c** HeLa cells were cotransfected with FLAG-tagged LAPTM5, 4A, 4B, or mock expression constructs along with Vpr-defective HIV-1 proviral vectors, as indicated. Two days after transfection, HIV-1 production was measured by p24 ELISA, and TZM-bl indicator cells were used to measure HIV-1 infectivity. Cells were lysed for western blotting to assess LAPTM5, 4A, 4B, gp120, and GAPDH expression. **P < 0.01 (two-tailed, unpaired Student's t-test), data are plotted as mean ± SEM of three independent experiments. All western blot data are representative of three independent experiments; their full-size images are presented in the Source Data.

to its degradation, thereby supporting our hypothesis that LAPTM5 functions to transport Env to the lysosome.

**LAPTM5 induces Env lysosomal degradation in macrophages.**
To explore whether Vpr counteracts the effect of endogenous LAPTM5 protein to release HIV-1 Env in macrophages, we used lentiviral shRNAs to silence LAPTM5 in primary MDMs (Fig. 6a) that were infected with equal amounts of wild-type or Vpr-defective HIV-1. The infected cell population were measured in this single-cycle infection system (Extended Data 9a), and we found that Vpr did not alter the level of gp120-containing virions in MDM culture in either low- or high-dose infection in the absence of LAPTM5, and the production of virion capsid was not affected (Fig. 6b, c). Vpr increased gp120 levels only in the presence of LAPTM5. These results indicate that Vpr promotes the levels of HIV-1 Env-containing virions by counteracting LAPTM5. Furthermore, to determine whether HIV-1 Env is transported to lysosomes by LAPTM5 for degradation by MDMs, we isolated lysosomes from MDMs infected with wild-type or Vpr-defective HIV-1. We administered raltegravir to inhibit subsequent rounds of infection at 2 days post infection to maintain a similar infection ratio of MDMs between wild-type and Vpr-defective HIV. Initially, the levels of Gag protein in the cell lysates were similar between wild-type and Vpr-defective HIV-1 (Fig. 6d), indicating similar infection ratios of MDMs. Silencing LAPTM5 appeared to restore Env protein production in cell lysates in the absence (lanes 1 vs 2, and 5 vs 6), but not in the presence (lanes 3 vs 4, and 7 vs 8) of the lysosomal inhibitor,

suggesting that LAPTM5-induced reduction of Env relies on lysosomes. Moreover, the presence of Vpr could reverse the LAPTM5-mediated reduction of Env (lanes 1 vs 5) in cell lysates when cells were not treated with the lysosome inhibitor BFLA. Accordingly, LAPTM5 protein was also decreased in the presence of Vpr (lanes 1 or 3 vs 5 or 7). In addition, Env was not detected within lysosomes in the presence of LAPTM5 without lysosomal inhibition (lanes 9, 13); conversely, lysosomal inhibition dramatically restored Env in lysosomal lysates in the presence of LAPTM5 (lanes 11, 15). Consistently, the presence of Vpr resulted in decreased levels of Env within lysosomes following treatment with the lysosome inhibitor (lanes 11 vs 15). In support, the LAPTM5 protein was also reduced in lysosomes in the presence of Vpr (lanes 11 vs 15). Collectively, LAPTM5 seems to transport Env to lysosomes, resulting in its degradation. If lysosome activities are uninhibited, the LAPTM5-translocated Env are already degraded within lysosomes, and thus cannot be detected within the lysosome (lanes 9 vs 11 and 13 vs 15).

Moreover, silencing LAPTM5 completely abolished the presence of Env in lysosomes even when a lysosome inhibitor was present (lanes 11 vs 12 and 15 vs 16), indicating that in the absence of LAPTM5, the Env protein cannot be translocated to lysosomes. Importantly, the presence of Vpr reduced the levels of LAPTM5 within lysosomes (lanes 11 vs 15), suggesting that Vpr promotes LAPTM5 degradation, thus reducing Env in the lysosomes undergoing LAPTM5 transport. In summary, these data collectively demonstrate that LAPTM5 may carry HIV-1 Env to the lysosomes for degradation in primary MDMs to restrict viral infection.

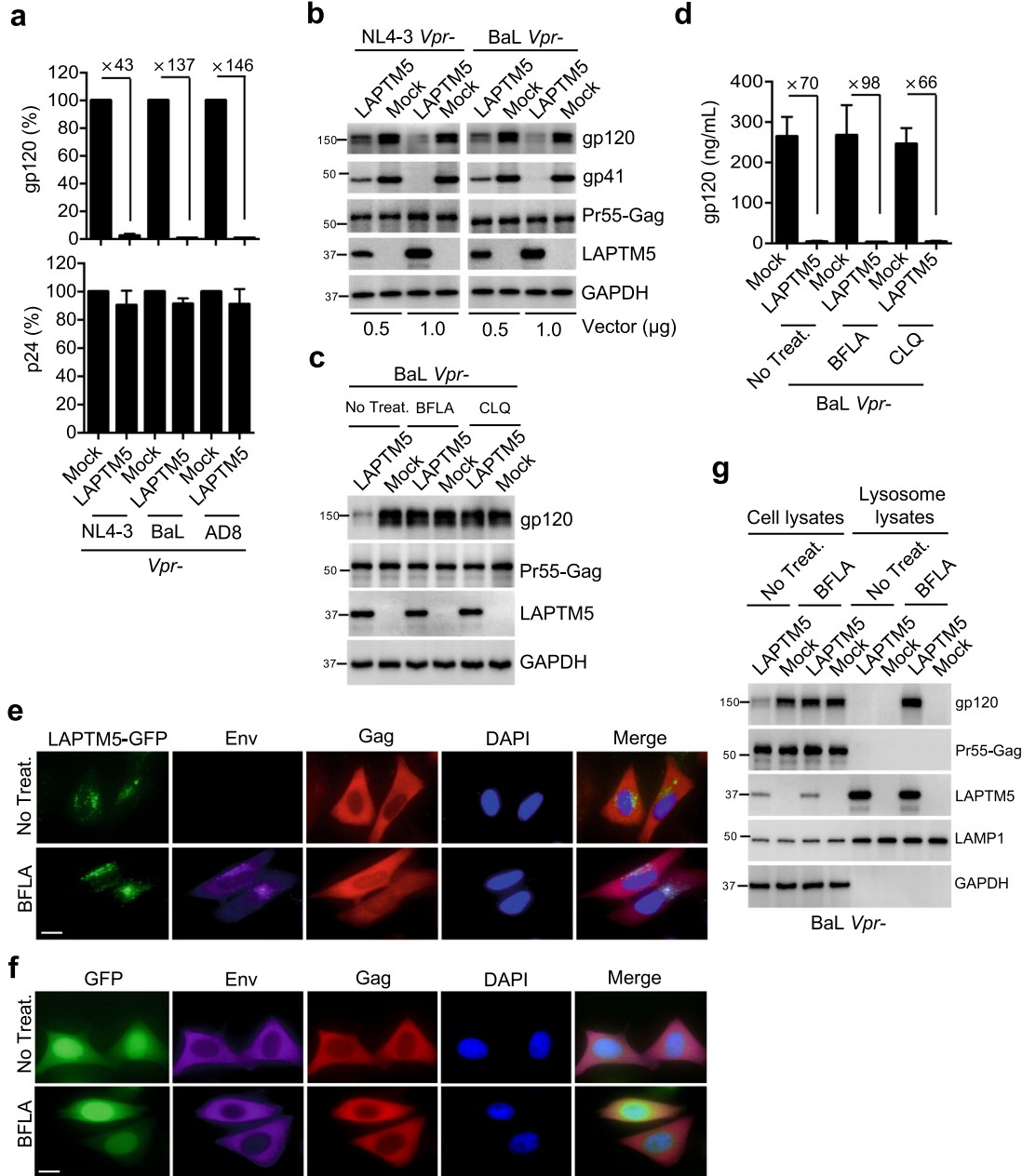

**Fig. 5 LAPTM5 transports HIV-1 Env to the lysosome for degradation. a** HeLa cells were cotransfected with FLAG-tagged LAPTM5 or mock expression constructs along with Vpr-defective HIV-1 proviral vectors. Two days after transfection, HIV-1 capsids or Env gp120-containing virions in culture were measured by p24 or gp120 ELISA, respectively. Data are plotted as mean ± SEM of three independent experiments. **b, c** HeLa cells were cotransfected with FLAG-tagged LAPTM5 or mock expression constructs along with Vpr-defective HIV-1 proviral vectors (**b**); or (**c**) in the absence or presence of lysosomal inhibitors (BFLA at 50 nM or CLQ at 30 μM treatment for the last 24 h). Two days after transfection, cells were lysed for western blotting to assess LAPTM5, Gag, gp120, gp41, and GAPDH expression. **d** HIV-1 Env gp120-containing virions in culture (in **c**) were measured by gp120 ELISA. Data are plotted as mean ± SEM of three independent experiments. **e, f** HeLa cells were cotransfected with GFP-tagged LAPTM5 (**e**) or mock (GFP) (**f**) expression constructs along with Vpr-defective HIV-1_NL4-3 in the presence or absence of a lysosomal inhibitor (50 nM). Two days after transfection, cells were fixed and immunostained with anti-gp120 or anti-p24 antibodies, which were probed with a second antibody conjugated with Alexa Fluor 647 or 555. Cell nuclei were stained with DAPI. Scale bars, 10 μm. Data are representative of three independent experiments. **g** HeLa cells were cotransfected with FLAG-tagged LAPTM5 or mock expression constructs along with Vpr-defective proviral vector of HIV-1_BaL in the presence or absence of a lysosomal inhibitor (BFLA at 50 nM treatment for the last 24 h). Two days after transfection, cells and isolated lysosomes were lysed for western blotting to assess LAMP1, LAPTM5, Gag, gp120, and GAPDH expression. All western blot data are representative of three independent experiments; their full-size images are presented in the Source Data.

**Vpr enhances HIV-1 infection in LAPTM5-expressing primary CD4+ T cells.** Vpr is not necessary for HIV-1 replication in many host cells—including 293T, HeLa, Jurkat cell lines, and primary CD4+ T cells—probably owing to their lack of LAPTM5 expression. To explore this, we introduced an LAPTM5 expression vector into TZM-bl cells (in which endogenous LAPTM5 protein was not detected) to produce LAPTM5 at similar levels as those of MDMs (Fig. 7a). We found that these LAPTM5 expression levels resulted in a requirement of Vpr for Env, but not capsid production, in TZM-bl cells (Fig. 7b, c, upper panel). Importantly, the

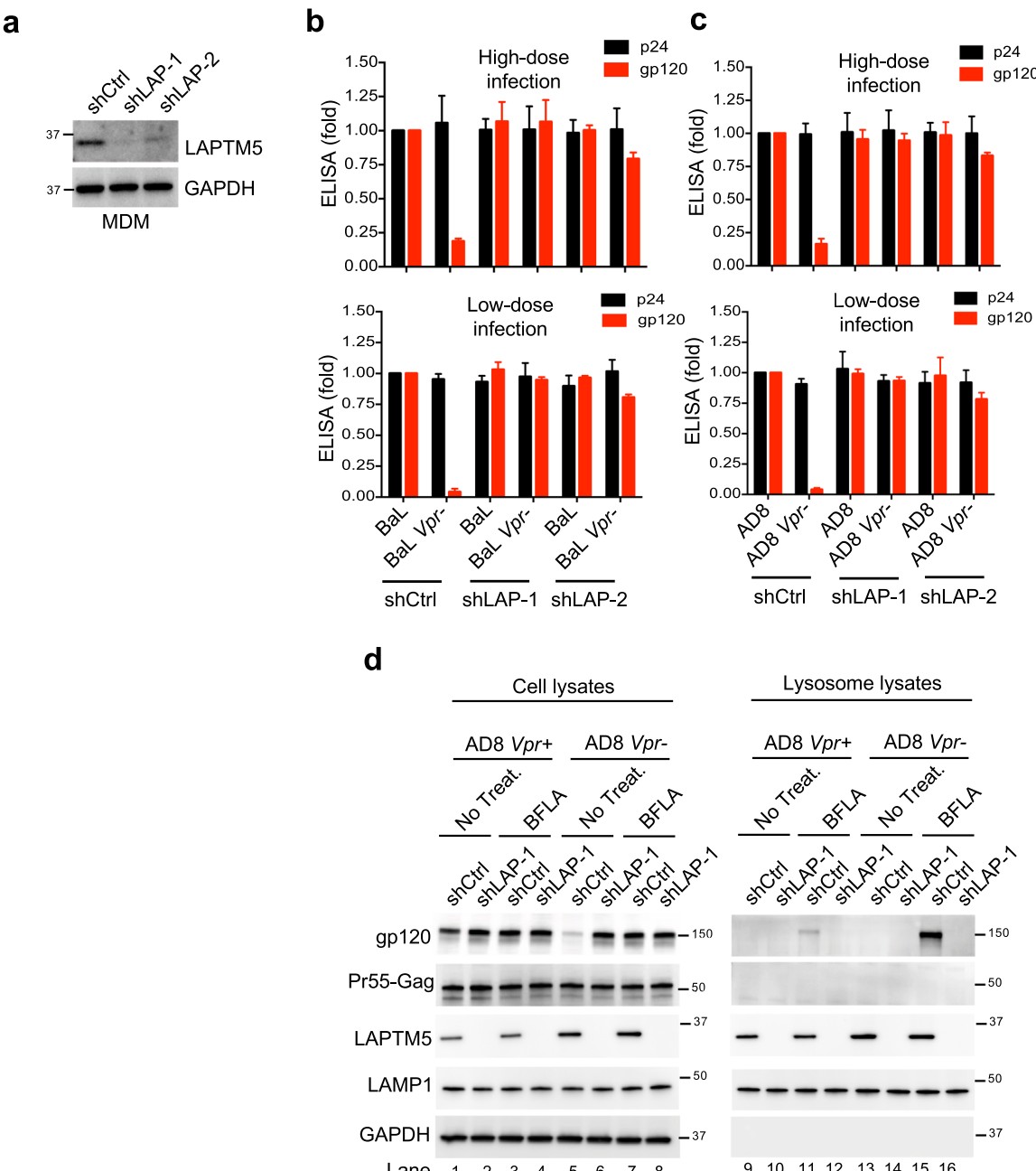

**Fig. 6 Vpr counteracts the LAPTM5-mediated degradation of HIV-1 Env. a–c** Lentiviral shRNA-transduced MDMs were infected with 10 or 100 ng of wild-type or Vpr-defective HIV-1 for 6 days. Before infection, the aliquoted shRNA-transduced MDMs were lysed for western blotting (**a**) to assess LAPTM5 and GAPDH expression. At 2 dpi, cells were treated with raltegravir (2 μM) to block subsequent rounds of infection. HIV-1 capsids or Env gp120-containing virions (**b**, **c**) in culture were measured by p24 or gp120 ELISA, respectively. Data are plotted as mean ± SEM of three independent experiments. **d** Lentiviral shRNA-transduced MDMs were infected with 500 ng of wild-type or Vpr-defective HIV-1$_{AD8}$ in the presence or absence of a lysosomal inhibitor (at 50 nM treatment for the last 24 h). At 2 dpi, cells were treated with raltegravir (2 μM) to block subsequent rounds of infection. Eight days after infection, cells and isolated lysosomes were lysed for western blotting to assess LAMP1, LAPTM5, Gag, gp120, and GAPDH expression. All western blot data are representative of three independent experiments; their full-size images are presented in the Source Data.

presence of Vpr also reduced LAPTM5 (lanes 1 vs 2 and 5 vs 6), whereas Vpr reversed the loss of Env in cell lysates (lanes 1 vs 2 and 5 vs 6). Moreover, similar levels of Gag protein in cell lysates indicated a comparable infection ratio between wild-type and Vpr-defective HIV-1 in this single-cycle infection. Therefore, these data suggest that Vpr overcomes the restriction activity of LAPTM5 to promote HIV-1 infection.

Furthermore, when CD4[+] T lymphocytes were transduced with a tetracycline-induced LAPTM5 expression lentiviral vector

so that they expressed levels of LAPTM5 similar to MDMs (Fig. 7d), this led to the dependence of HIV-1 on Vpr for infecting other CD4[+] T cells (Fig. 7e, the infected cell population as shown in Supplementary Fig. 9b). In the absence of LAPTM5, Vpr had no obvious effects. Importantly, the similar levels of p24 in the absence of LAPTM5 (Fig. 7e, lower panel) indicate that wild-type and Vpr-defective HIV-1 at least exhibited a similar infection ratio; thus, the significant difference between wild-type and Vpr-defective 89.6 in the presence of LAPTM5 (Fig. 7e, lower

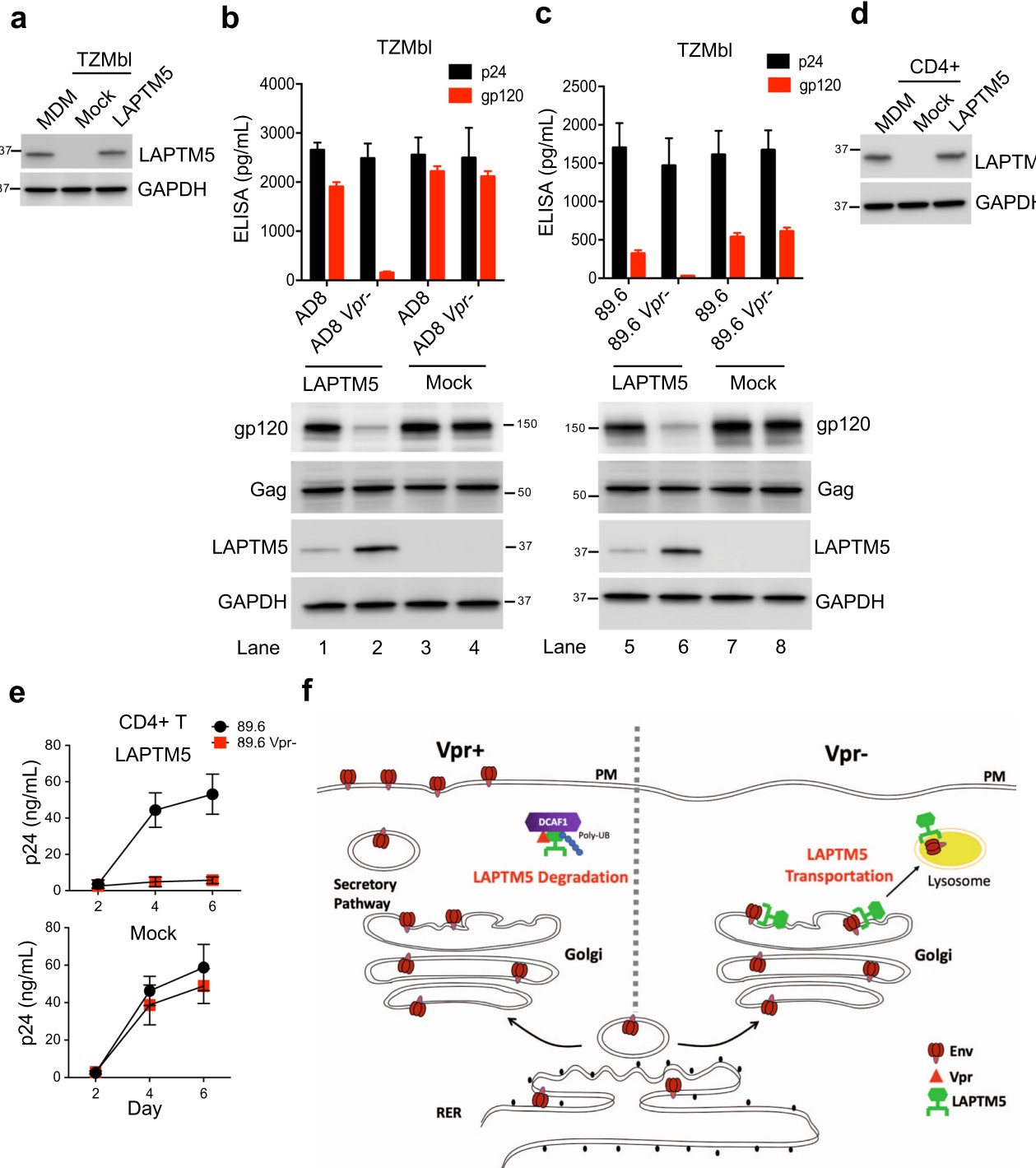

**Fig. 7 Vpr enhances HIV-1 infection in LAPTM5-expressing CD4+ T cells. a–c** TZM-bl cells were transfected with FLAG-tagged LAPTM5 or mock expression constructs to express LAPTM5 at similar levels as in MDMs. The same number of TZM-bl cells or MDMs was lysed for western blotting to assess LAPTM5 and GAPDH expression (**a**). LAPTM5-positive or -negative-TZM-bl cells were infected with 100 ng of wild-type or Vpr-defective HIV-1 (**b, c**) for 3 days. At 1 dpi, cells were treated with raltegravir (2 μM) to block subsequent rounds of infection. HIV-1 capsids or Env-containing virions in culture were measured by p24 or gp120 ELISA, respectively. Data are plotted as mean ± SEM of three independent experiments. Cells were lysed for western blotting to assess gp120, Gag, LAPTM5 and GAPDH expression. **d, e** CD4+ T cells were stimulated and transduced with tetracycline-controlled LAPTM5 or mock expression lentiviral vectors to produce LAPTM5 at similar levels as in MDMs. The same number of CD4+ T cells or MDMs was lysed for western blotting to assess LAPTM5 and GAPDH expression (**d**). LAPTM5-positive or -negative-CD4+ T cells were infected with 100 ng of dual-trophic wild-type or Vpr-defective HIV-1$_{89.6}$ for 6 days (**e**). Viral production was measured by p24 ELISA at the indicated time points. **f** Working model of how Vpr counteracts LAPTM5 to promote HIV-1 infection. Data are plotted as mean ± SEM of three independent experiments. All western blot data are representative of three independent experiments; their full-size images are presented in the Source Data.

panel) is likely due to LAPTM5 expression and not their infection ratios. Therefore, the lack of LAPTM5 is likely to be the reason why Vpr is dispensable for HIV-1 infection of primary CD4+ T cells. For this reason, we proposed a theoretical model of how Vpr counteraction of LAPTM5 enhances HIV-1 infection in macrophages (Fig. 7f). LAPTM5 is known to target and transport cargo from the Golgi complex to lysosomes[27–29]. In the absence of Vpr, HIV-1 Env is synthesized as a polyprotein precursor on the rough endoplasmic reticulum and transported to the Golgi complex for oligosaccharide processing, where LAPTM5 hijacks Env for lysosomal degradation. The presence of Vpr enhances HIV-1 infection by promoting LAPTM5 degradation via the DCAF1/proteasome, thus releasing Env from the Golgi complex traffic to the plasma membrane for HIV-1 progeny assembly. To make this model succinct, we have not included HIV-1 assembly in the virus-containing compartments (VCCs) of macrophages, and VCCs probably represent specialized, intracellularly sequestered portions of the plasma membrane[33–36], where viral progeny assemble in the majority of host cells.

## Discussion

In this study, we demonstrated that LAPTM5 is responsible for the anti-HIV-1 activity in macrophages, which is suppressed in the presence of Vpr. In agreement, we observed that Vpr had no obvious effect on viral capsid production in the first round of infection, whereas it enhanced HIV-1 Env levels to promote progeny virion infectivity[26]. However, EFV or raltegravir treatment completely abolished LAPTM5 degradation, indicating that the degradation of LAPTM5 is dependent on viral productive infection. Moreover, HIV-1 virion-packaged Vpr from producer cells did not promote efficient LAPTM5 degradation in MDMs, possibly because of insufficient packaged Vpr proteins or the packaged Vpr cannot be stored for long period in macrophages to promote LAPTM5 degradation. Therefore, the degradation of LAPTM5 may be dependent on the newly synthesized Vpr during HIV-1 infection. Furthermore, we observed that Vpr could promote LAPTM5 polyubiquitination to degrade it in a DCAF1-dependent manner. In the absence of DCAF1, LAPTM5 ubiquitination and subsequent degradation induced by Vpr was inhibited. Moreover, LAPTM5 expression is present in MDMs, MDDCs, and monocytes but not in various cell lines, such as 293T, HeLa, and Jurkat, as well as a trace amount of LAPTM5 transcripts found in stimulated CD4+ T cells, suggesting that Vpr is not necessary for HIV-1 replication in these cell lines. Interestingly, we also observed a similar expression of LAPTM5 in resting CD4+ T cells in contrast to MDMs. Therefore, whether LAPTM5 limits HIV-1 Env protein in resting CD4+ T lymphocyctes remains to be solved.

Moreover, we found evidence that LAPTM5 can target HIV-2 and the primate lentiviruses SIV$_{mac239}$ and SIV$_{agm}$ but not MLV and FIV to restrict host infection, raising the possibility that LAPTM5 has a broader role in innate antiviral immunity against primate lentiviruses. Interestingly, HIV-2 infectivity was shown to be much lower than that of HIV-1 when the virus was produced in MDMs[37], and Gea-Mallorquí et al. recently reported that HIV-2 infection of macrophages produces and accumulates poorly infectious viral particles[38]. It is likely that the presence of LAPTM5 is partly responsible for the poorly infectious HIV-2 virions produced in macrophages, given that Vpr of HIV-2 cannot promote degradation of human LAPTM5. Nevertheless, further studies are required to evaluate the endogenous LAPTM5 inhibition of HIV-2 in macrophages. In summary, we have confirmed that Vpr enhances the spread of HIV-1 in macrophages by counteracting LAPTM5 that translocates Env to the lysosome for degradation, thus impairing the release of Env-

containing virions. This pathway is dependent on the expression of the Vpr cofactor DCAF1.

Interestingly, during this paper revision, Lubow et al. also reported that Vpr could synergistically with Nef target distinct stages of mannose receptor (MR) biosynthetic pathway and reduce MR protein expression[39]. Notably, Nef had been shown to mediate post-translational downregulation and redistribution of MR to increase HIV-1 infection[40], and Lubow et al. further showed in their paper that Vpr specifically reduced MR transcription and Nef inhibited MR expression on cell surface, and both of them synergistically to counteract MR protein-mediated suppression of HIV-1 replication in macrophages[39]. Together with our findings, it is necessary in the future research to elucidate whether LAPTM5 is involved in the MR pathway to reduce Env that Vpr counteracts. Taken together, in our study, we have demonstrated how HIV-1 evades innate immunity via Vpr to counteract the antiviral effects of LAPTM5, which is responsible for the anti-HIV-1 activity in macrophages. The ectopic expression of LAPTM5 and LAPTM4A potently inhibited wild-type HIV-1, even in the presence of Vpr, demonstrating that these antiviral factors could be exploited as anti-HIV-1 therapeutic agents.

## Methods

**Ethics statement**. The Research and Ethics Committee of The First Affiliated Hospital of China Medical University approved the study. All blood samples isolated from healthy donors provided written informed consent following the National Health and Medical Research Council guidelines. Informed consent was obtained from each healthy donor before the study. The Institutional Review Board approved the study protocol and informed consent forms of China Medical University.

**Cells and culture reagents**. HeLa (ATCC, Cat.CCL-2), 293T (ATCC, Cat.CRL-11268), mouse NIH3T3 cells (NIH AIDS Reagent Program, Cat.ARP-9946), TZM-bl (provided by Dr Guangxia Gao)[41], and Jurkat (ATCC,Cat.TIB-152) cells were grown and maintained in Dulbecco's modified Eagle's medium (Gibco) or RPMI-1640 medium (Gibco), as previously described[42]. Both media were supplemented with 10% fetal bovine serum (FBS, Gibco), 100 U/mL penicillin, and 100 mg/mL streptomycin. Plasmids were transfected into 293T cells using Lipofectamine 2000 (Invitrogen) according to the manufacturer's instructions. MOLT-4 (ATCC, Cat. CRL-1582) cells were grown and maintained in RPMI-1640 medium modified to contain 2 mM L-glutamine, 10 mM HEPES, 1 mM sodium pyruvate, 4500 mg/L glucose, and 1500 mg/L sodium bicarbonate (Gibco)[43]. PBMCs obtained from healthy blood donors were purified using Ficoll-Hypaque gradient centrifugation. CD4+ T cells or monocytes were isolated from PBMCs via negative selection with human CD4+ T cells or CD14-positive enrichment cocktail (StemCell Technologies). To stimulate CD4+ T cells, CD3/CD28 activator magnetic beads (Invitrogen) were added to the culture medium and incubated for 2 days along with IL-2 (50 U/mL; Biomol) according to the manufacturer's instructions. The isolation and culture of monocytes, MDMs, and MDDCs was performed as previously described[42]. Briefly, MDMs were generated by stimulating monocytes with 50 ng/mL recombinant human granulocyte-macrophage colony-stimulating factor (GM-CSF; R&D) for 7 days. MDDCs were generated by incubating CD14-purified monocytes in IMDM medium (Gibco) supplemented with 10% FBS, 2 mM L-glutamine, 100 IU/mL penicillin, 100 mg/mL streptomycin, 10 mM HEPES, 1% nonessential amino acids, 1 mM sodium pyruvate, 10 ng/mL GM-CSF, and 50 ng/mL IL-4 (Miltenyi Biotec). On day 4, two-third of the culture medium was replaced with fresh medium containing GM-CSF and IL-4. Immature MDDCs were harvested and used for experiments on day 6.

**Lysosome isolation**. Lysosomes were isolated from HeLa cells or MDMs according to the manufacturer's instructions (Lysosome Enrichment Kit for Tissue and Culture Cells, Thermo). Briefly, cells were pelleted (50–200 mg) by centrifuging harvested cell suspensions at 850 ×g for 2 min. The supernatants were removed, and the resulting cell pellets were resuspended by adding lysosome enrichment reagent A on ice. The cell suspension was sonicated for cell lysis (e.g., a 10–15 s burst at 6–9 W) and subsequently treated with lysosome enrichment reagent B. In an ultracentrifuge tube, a discontinuous density gradient was prepared by carefully overlying the prepared OptiPrep gradient in descending concentrations. The samples were ultracentrifuged at 145,000 ×g for 2 h at 4 °C. After centrifugation, the lysosomal band was harvested at the top of the gradient and mixed with 2–3 volumes of PBS to decrease the concentration of the OptiPrep Media. The mixtures were centrifuged at 18,000 ×g for 30 min at 4 °C. The lysosome pellets were lysed for western blotting analysis.

**Chemical reagents**. MG132, chloroquine (CLQ), and bafilomycin A1 (BFLA) were purchased from Sigma Aldrich.

**Plasmids**. The LAPTM5, 4A, and 4B expression vectors were purchased from OriGene, or their open reading frames were de novo cloned into pCMV-3Tag-2A (Addgene) vector. The LAPTM5 open reading frame was cloned into pLenti-TRE-rtTA3-IRES-Puro lentiviral vectors (Addgene), in which LAPTM5 expression was controlled by the presence of tetracycline. The LAPTM5 open reading frame with an IRES-GFP cassette was cloned into a CMV-driven pcDNA3.1 vector (Invitrogen). HIV-1 reporter vectors of NL4-3.Luc.R-E-, NL4-3.GFP.R−E− and proviral vectors of NL4-3-R-E-, NL4-3, 89.6, SIV$_{mac239}$, SIV$_{agm}$, HIV-2$_{Rod}$, FIV-14-Peta-luma, and MLV were obtained from the NIH AIDS Program. HIV-1 proviral vectors of BaL Vpr$^{+/−}$ were gifted by Dr N. R. Landau[11]; pNL-AD8 was gifted by Dr E. Freed; pNL-AD8 Vpr$^−$ was gifted by Dr Y. Zheng[44]; and pHIV-89.6 Vpr$^−$ was gifted by Dr G. Gao. Recombinant lentiviruses for the microRNA-adapted pGIPZ-shRNAs (Catalog numbers #37067 and #37068 targeting the open reading frame or #84597 targeting the 3′-UTR of LAPTM5 or a control, #RHS4346) were generated via the transient transfection of lentiviral pGIPZ-shRNA vectors and trans-lentiviral shRNA packing packaging plasmids (pTLA1-PAK, pTLA1-ENZ, pTLA1-ENV, pTLA1-TOFF, and pTLA1-TAT/REV) in 293T cells according to the manufacturer's instructions (Dharmacon). Vpr derived from different virus isolates NL4-3, 89.6, AD8, BaL, Yu2, HIV-2$_{Rod}$, and SIV$_{mac239}$ were cloned into the pCMV-3Tag-2A (Addgene) vector.

**RNA interference in THP-1 cells, MDMs**. To achieve shRNA-mediated silencing of *LAPTM5*, a microRNA-adapted shRNA lentivirus was introduced into THP-1 cells and MDMs. Briefly, the freshly isolated monocytes were treated with VLP-Vpx and transduced with shRNA lentivirus particles[26,45,46]. After puromycin selection, the cells were infected with replication-competent HIV-1 and washed twice with cold PBS 6 h after infection to remove the input virus. Stealth-grade siRNA (HSS111754, HSS111755, HSS111756) against *LAPTM5* and controls were purchased from Invitrogen. To achieve siRNA-mediated silencing, differentiated THP-1 cells were directly transfected with siRNA using Lipofectamine 3000 (Thermo Fisher).

**ELISA detection of virions**. Levels of HIV-1 virion containing p24 or subtype B gp120 proteins in culture supernatants were measured by ELISA according to the manufacturer's instructions (ABL Corporation).

**Measurement of viral infectivity**. The HIV-1, HIV-2, and SIV produced in culture supernatants were measured by infecting TZM-bl reporter cells, and the background luminescence units from the control TZM-bl cells were subtracted from each data point.

**Measurement of late reverse-transcript products of FIV and MLV**. MOLT-4 cells or mouse NIH3T3 were infected with FIV or MLV for 96 or 24 h, respectively, and genomic DNA was extracted for qPCR analysis of the late reverse-transcript product, using specific primers in Supplementary Table 2, as previously described[47–49].

**Luciferase detection assay**. Luciferase activity was quantified as relative luminescence units in the cell lysates according to the manufacturer's instructions (Promega).

**Identification of LAPTM5 interaction with Vpr in MDMs**. MDMs generated from the monocytes isolated from ten healthy donors were pretreated with VLP-Vpx and transduced with a lentiviral vector expressing FLAG-Vpr from NL4-3. After puromycin selection, cells ($5 \times 10^6$) were treated with or without 1.5 μM MG132 for 8–10 h and lysed using IP lysis buffer (50 mM Tris-HCl, pH 7.2, 50 mM NaCl, 1% NP-40, 1 mM EDTA, 2% glycerol, 1× protease inhibitor cocktails, cOmplete), and the cell lysates from five donors were pooled (then two groups were obtained from a total of ten donors for subsequent experiments). The cell lysates were incubated on ice for 30 min and centrifuged at 14,000 ×g for 10 min at 4 °C. The supernatants were transferred to fresh tubes, and the pellets were mixed with cold IP lysis buffer before sonication, followed by a second round of centrifugation at 14,000 ×g for 10 min at 4 °C. The supernatants obtained from the two extraction steps were pooled and incubated with a mix of protein A and protein G Dynabeads (Invitrogen), which had been pretreated overnight with an anti-FLAG antibody at 4 °C. The IP products were washed with cold IP buffer and PBST 5–10 times (500 μL per wash). The IP products were first reduced in 20 mM dithiothreitol (Sigma) at 95 °C for 5 min and subsequently alkylated in 50 mM iodoacetamide (Sigma) for 30 min in the dark at room temperature. After alkylation, the samples were transferred to a 10 kDa centrifugal spin filter (Millipore) and, sequentially, washed with 200 μL of 8 M urea three times and 200 μL of 50 mM ammonium bicarbonate twice via centrifugation at 14,000 ×g. Next, tryptic digestion was performed by adding trypsin (Promega) at a ratio of 1:50 (enzyme/substrate, m/m) in 200 μL of 50 mM ammonium bicarbonate at 37 °C for 16 h. Peptides were recovered by transferring the filter to a new collection tube and centrifuging at 14,000 ×g.

To increase the yield of peptides, the filter was washed twice with 100 μL of 50 mM NaHCO$_3$. Peptides were desalted using a StageTip. MS experiments were performed on a nanoscale UHPLC system (EASY-nLC1000 from Proxeon Biosystems, Odense, Denmark) connected to an Orbitrap Q-Exactive equipped with a nano-electrospray source (Thermo Fisher Scientific, Bremen, Germany). The peptides were dissolved in 0.1% FA with 5% CH$_3$CN and separated on an RP-HPLC analytical column (75 μm × 15 cm) packed with 2-m C18 beads (Thermo Fisher Scientific) using a 2-h gradient ranging from 5 to 40% acetonitrile in 0.5% formic acid at a flow rate of 250 nL/min. The spray voltage was set to 2.5 kV, and the temperature of the ion transfer capillary was 275 °C. A full MS/MS cycle consisted of one full MS scan (resolution, 70,000; automatic gain control [AGC] value, 1e6; maximum injection time, 50 ms) in the profile mode over a mass range of m/z 300 and 1800, followed by fragmentation of the ten most intense ions via high-energy collisional dissociation with normalized collision energy at 28% in the centroid mode (resolution, 17,500; AGC value, 1e5; maximum injection time, 100 ms). Moreover, the dynamic exclusion window was set to 40 s, and one microscan was acquired for each MS and MS/MS scan. Subsequently, unassigned ions and those with a charge of 1+ and >7+ were rejected for MS/MS, and lock mass correction using a background ion (m/z 445.12003) was applied[50]. The raw data were processed using Proteome Discoverer (PD, version 2.1), and MS/MS spectra were used to search the reviewed Swiss-Prot human proteome database. All searches were performed with a precursor mass tolerance of 7 ppm and a fragment mass tolerance of 20 millimass units, with oxidation (Met) (+15.9949 Da) and acetylation (protein N-termini) (+42.0106 Da) as variable modifications, carbamidomethylation (+57.0215 Da) as the fixed modification and two trypsin-missed degradations were allowed. Only peptides of at least six amino acids in length were considered. The peptide and protein identifications were filtered by PD to control the false discovery rate at <1%. At least one unique peptide was required for protein identification.

**Co-immunoprecipitation**. HeLa cells ($5.0 \times 10^6$) were lysed with IP lysis buffer (50 mM Tris-HCl, pH 7.2, 50 mM NaCl, 1% NP-40, 1 mM EDTA, 2% glycerol, and 1× protease inhibitor cocktails, cOmplete). The lysates were incubated on ice for 30 min and centrifuged at 14,000 ×g for 10 min at 4 °C. The supernatants were transferred to fresh tubes, and the pellets were mixed with cold IP lysis buffer before sonication, followed by a second round of centrifugation at 14,000 ×g for 10 min at 4 °C. The supernatants obtained from the two extraction steps were pooled and incubated with a mix of protein A and G Dynabeads (Invitrogen), which had been pretreated overnight with antibody at 4 °C. The IP products were washed with cold IP and PBST buffers 5–10 times (500 μL per wash).

**Western blotting and antibodies**. Western blotting was performed using the standard method to detect cellular proteins. The antibodies used in this study were as follows: polyclonal rabbit anti-LAPTM5 (Biorbyt, Cat. orb184851,1:1000), monoclonal mouse anti-LAMP1 (Abcam, Cat.ab25630, 1:1000), polyclonal goat anti-gp120 (NIH AIDS Reagent Program, Cat.ARP-288, 1:20000), human monoclonal anti-gp41 (NIH AIDS Reagent Program, Cat.ARP-11557, 1:5000), rabbit anti-GAPDH (Thermo, Cat.PA1-987, 1:1000), mouse monoclonal anti-FLAG (SIGMA, Cat.F1804, 1:1000), rabbit polyclonal anti-p24 (Abcam, Cat.ab63913, 1:1000), mouse monoclonal anti-GFP (Abmart, Cat.M20004, 1:2000), mouse monoclonal anti-HA (Abmart, Cat.M20003, 1:2000), mouse Igs-HRP (Abcam, Cat.6789, 1:5000), Higly cross-Adsorbed Alexa Fluor 488-, 647-, or 555-labeled goat anti-rabbit secondary antibody (Thermo, Cat.A32731, 1:200; Cat.A32733; 1:200; Cat.A32732, 1:200), Highly cross-Adsorbed Alexa Fluor 488-, 647-, or 555-labeled goat anti-mouse secondary antibody (Thermo, Cat.A32723, 1:200; Cat. A32728; 1:200; Cat.A32727, 1:200), rabbit and mouse IgG Trueblot (eBioscience, Cat.18-8816-33,1:1000; Cat.18-8817-33, 1:1000), and rabbit or mouse IgG isotype control (Abcam, Cat.18413; Cat.ab18413).

**Microscopy**. Cells were overlaid on poly-L-lysine-coated glass slides, and the images were collected using the Thermo Fisher EVOS™ FL Imaging System. HeLa, MDMs, or MDDCs were fixed and stained with anti-LAPTM5, anti-LAMP1, anti-gp120, or anti-p24, which were probed with an Alexa Fluor 488-, 555- or 674-conjugated goat anti-rabbit or anti-mouse secondary antibody according to the manufacturer's instructions (Imag-iT Fixation/Permeabilization Kit, Invitrogen).

**qPCR**. Total RNA was extracted from cells using TRIzol (Invitrogen) according to the manufacturer's instructions. The obtained RNA was dissolved in 100 μL of DPEC-H$_2$O, and 1 μg of the purified RNA was treated with DNase I (amplification grade, Invitrogen) for 10–15 min at room temperature according to the manufacturer's instructions. Thereafter, RNA was immediately primed with oligo-dT and reverse-transcribed using Superscript III Reverse Transcriptase (Invitrogen). Real-time PCR analysis was performed using the ΔΔCT method[42]. The results were normalized against the amplification results for the internal control (GAPDH). The primers used in this study are shown in Supplementary Table 2.

**Statistical analysis**. Statistical analysis was performed using Prism 6.0 (Graph Pad Software) and unpaired, two-tailed Student's t-tests were used for statistical comparison between groups, unless otherwise specifically mentioned.

**Reporting summary**. Further information on research design is available in the Nature Research Reporting Summary linked to this article.

## Data availability

A reporting summary for this article is available as a Supplementary Information file. Liquid chromatography and mass spectrometry data are available in Mendeley Data (https://data.mendeley.com/datasets/62tgy3jbtg/1). Data supporting the findings of this study are available within the article and its Supplementary Information files and also from the corresponding authors upon reasonable request. Source Data are provided with this paper.

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

## Acknowledgements

We thank Dr Nathaniel Landau for gifting BaL $Vpr^{+/-}$ vectors, Dr Guangxia Gao for gifting the HIV-1$_{89.6}$ $Vpr^-$ vector, Dr Eric Freed for gifting the HIV-1$_{AD8}$ vector, and Dr Yonghui Zheng for gifting the HIV-1$_{AD8}$ $Vpr^-$ vector. We thank all our laboratory members for contributing to this study. We would also like to thank Jingjing Chen for technical support and Xiaotian Ma for critical reading and editing of our manuscript. This study was supported by the Mega-projects of National Science Research for the 13th Five-Year Plan (Grant No. 2017ZX10201101; Grant No. 2017ZX09304025), and the National Natural Science Foundation of China (Grant No. 82072284).

## Author contributions

G.L. directed and conceived the research. L.Z., S.W., M.X., Y.H., X.Z., Y.X., H. Sun, H.D., and W.G. performed the experiments or analyzed the data. H. Shang and H.D. provided intellectual advice on experimental design. G.L. wrote the manuscript.

## Competing interests

The authors declare no competing interests.
