## [Peer Review File · Nature Communications]

REVIEWER COMMENTS

Reviewer #2 (Remarks to the Author):

In this new version of their manuscript, the authors have answered all my concerns. I'm satisfied with their corrections and I recommend publication.

Reviewer #5 (Remarks to the Author):

This is a beautiful study that makes a convincing case that LAPTM5 is a cellular factor responsible for the restriction of Vpr-null virus in infected MDM. I am happy with the responses to the previous critiques by reviewer #1. I have a few additional relatively minor issues that the authors might want to consider.

Line 98: the statement that Vpr cannot promote infection of HeLa cells is trivial since these cells are CD4(-) and are therefore resistant to infection, period. This has nothing to do with Vpr. Unless the authors have a specific reason to make that point, I would suggest to just delete it.

Line 116-117: I agree with the authors that virion associated Vpr is unlikely to be stable for 6 days following infection of target cells. Therefore, this experiment does not prove that virion associated Vpr is less capable of degrading LAPTM5 than de novo synthesized intracellular Vpr. Since this issue is not really critical for the overall take-home message of this study, I would recommend to simply delete Extended Fig. 1c and the corresponding text in the manuscript (lines 113 to 117).

In Extended Figure 3B/3C the authors find that Vpr from HIV-2 Rod neither degrades human nor macLAPTM5. Also, in Extended Figure 6b, the authors demonstrate restriction of HIV-2Rod by LAPTM5. So how does HIV-2 manage to replicate in LAPTM5 expressing human cells? The authors should offer a plausible explanation for this conundrum.

Reviewer #6 (Remarks to the Author):

Comment 1: Reviewer 3 questions whether the localization of Vpr (primarily nuclear) makes sense given its effects on LapTM5, which is cytoplasmic. The authors respond with a literature review and extended data Figure 5. While the authors succeed in demonstrating that the localization of Vpr is complex and not well-defined, the overarching concern is not well-addressed. In addition to Vpr being nuclear, its cellular cofactors (E3-ubiquitin ligase CUL4A-DDB1 DCAF) and other reported targets have all been nuclear. Thus, the concern remains.

Comment 2: Reviewer 3 expresses concern about the surprising result that the replication curves for two different viruses match so closely point by point in figure 1D and extended figure 1. We agree that the curves plus knockdown are nearly perfectly overlaid, which does not seem likely based on experience with primary macrophages, which are famously heterogeneous in their responses. Moreover, Donors 3, 12 and 13 are shown but what about donors 1, 2, 4,5,7,8,9,etc.? This pattern is concerning. Was only a subset of selected data shown? The response of the authors was that they provided additional data for donors 12-15 but extended data Figure 1 only shows donors 6, 14, 15 and 16. These additional examples share the same appearance where the curves with knockdown are nearly perfectly overlaid. The additionally data provided for the reviewers showing that a 50% knockdown has little to no significant effect doesn't reassure given the apparent selectivity with which the authors are revealing information.

Comment 3: Reviewer 3 is also concerned about how the results in this paper fit with the literature regarding other restriction factors reported to be reversed by Vpr. The authors responded by suggesting that LAPTM5 could be a central molecule in the previously reported mannose receptor-mediated restriction of HIV Env but they provide no evidence to support this assertion.

Comment 4: Reviewer 3 is concerned about prior reports that LAPTM5 is expressed in CD4+ T cells. The author's response, that T cell activation downmodulates LAPTM5 appears to be sufficient to alleviate the concern.

Comment 5: Reviewer 3 is concerned that HIV-2 Vpr does not degrade LAPTM5 as HIV-2 would also need to get around the same restriction. The authors speculate that HIV-2 may have another way to get around this restriction. While this is not very satisfying, it is conceivable and probably beyond the scope of this paper.

Additional Comments- Many controls are missing and some data are concerning:

1. Figure 1 and extended Figure 1 are problematic with respect to the conclusion that decline in LAPTM5 is Vpr-dependent. In the MDM system, the proportion of cells infected by WT virus is much greater than that by the Vpr-mutant. There is no control for these drastic differences in proportion of infected cells. This would be made obvious by including Gag on the western blot or including flow cytometric assessment of the proportion of cells that were infected based on intracellular Gag stain. Thus, the alternative hypothesis that Vpr affects LAPTM5 indirectly by increasing the infection rate is not excluded.

2. Figures 2 and 3 as well as their extended figures show overexpression studies using HeLa cells in which they claim LAPTM5 decreases virion infectivity. The results show decreased viral infection following treatment of an indicator line with HeLa cell supernatant plus and minus HeLa cell transfection with LAPTM5. However the controls are insufficient to conclude that LAPTM5 decreases virion infectivity. Based on the data shown, the results could either be due to decreased virion release or a decrease in per particle infectivity. This could be resolved by including the p24 ELISA/Env ELISA shown in later figures. Figure 4 is better, but still lacks Env data.

3. Extended figure 2c shows LAPTM5 expression in THP1 cells. It is curious that no experiments were done in this cell line that naturally expresses LAPTM5.

4. Figure 3 Part e has the same problem as for Figure 1 – however this figure is better because the authors are now showing Gag expression. This makes it clear that Gag expression is clearly lower in Vpr negative infected MDMs as you would expect given the expected lower proportion of infected cells in the Vpr-negative infection condition. Based on these results, one can't conclude reduction of LAPTM45 is due to Vpr directly. It could be playing a role indirectly through increasing the proportion of infected cells in the sample.

5. Figure 5 Shows over expression of LAPTM5 in HeLa cells leads to loss of Env but does not show this is reversible by Vpr. Everything is in a Vpr-negative background. No evidence that LAPTM5-dependent targeting of Env to lysosomes is reversed by Vpr in part g. This is really important to show.

6. Figure 6 infection rates are missing- what proportion of the single round infections were Gag+? This is important because Vpr-dependent restriction factors can be overcome by high multiplicity of infection. Does knockdown of LAPTM5 increase proportion of infected cells in single round infections shown in Figures 6b and C? Figure 6d should include data demonstrating that targeting of Env to lysosomes is reversible by Vpr. This is really important to show.

7. Figure 7 Controls and obvious experimental results are missing - Need to round out the story by demonstrating that LAPTM5 targets Env for degradation and that this is reversible by Vpr for BOTH cell types. Key missing controls includes the proportion of infected cells for each experiment, which can easily be done with intracellular Gag staining.

8. I am concerned about the western blots shown in Figure 7a and 7d (full blots are shown in file named 260399_1_data_set_4992922_qhx6sf.pdf). It doesn't make sense that mock treated TZMBL and primary T cells, which are very different kinds of cells have the same cross-reacting lower background band and that in both cases this same background band mysteriously goes away in LAPTM5 transfected/transduced cells. Given the pattern shown, I am suspicious that macrophage lysate was substituted for the transduced/transfected cells and mislabeled in the figure.

Response to Reviewers' Comments

We thank the reviewers for their helpful comments and suggestions, which have helped us significantly improve the manuscript. All changes are marked **in red** in the revised manuscript, and our point-by-point responses to the reviewers' comments are presented below.

Reviewer #2 (Remarks to the Author):

In this new version of their manuscript, the authors have answered all my concerns. I'm satisfied with their corrections and I recommend publication.

Response 1: Thank you for the helpful comments that have helped us improve our manuscript.

Reviewer #5 (Remarks to the Author):

This is a beautiful study that makes a convincing case that LPTM5 is a cellular factor responsible for the restriction of Vpr-null virus in infected MDM. I am happy with the responses to the previous critiques by reviewer #1. I have a few additional relatively minor issues that the authors might want to consider.

Line 98: the statement that Vpr cannot promote infection of HeLa cells is trivial since these cells are CD4(-) and are therefore resistant to infection, period. This has nothing to do with Vpr. Unless the authors have a specific reason to make that point, I would suggest to just delete it.

Response 1: Thank you for this suggestion. We have deleted the phrase (*.....whereas it cannot promote infection in HeLa cells.*) in this revision.

Line 116-117: I agree with the authors that virion associated Vpr is unlikely to be stable for 6 days following infection of target cells. Therefore, this experiment does not prove that virion associated Vpr is less capable of degrading LPTM5 than de novo synthesized intracellular Vpr. Since this issue is not really critical for the overall take-home message of this study, I would recommend to simply delete Extended Fig. 1c and the corresponding text in the manuscript (lines 113 to 117).

Response 2: Thank you; we have made the necessary changes.

In Extended Figure 3B/3C the authors find that Vpr from HIV-2 Rod neither degrades human nor macLPTM5. Also, in Extended Figure 6b, the authors demonstrate restriction of HIV-2Rod by LPTM5. So how does HIV-2 manage to replicate in LPTM5 expressing human cells? The authors should offer a plausible explanation for this conundrum.

Response 3: Thank you. Following your suggestion, we have added the following explanation in the Discussion section (line 352-358, page 12) to explain why HIV-2 can replicate in macrophages but its viral particle has lower infectivity:

In contrast to HIV-1, only a small number of studies have evaluated HIV-2 infection in macrophages. We propose that because LPTM5 targets Env and HIV-2's Vpr does not induce degradation of LPTM5. HIV-2 could efficiently replicate in macrophages to produce virion progenies in a single-cycle infection, but with poorly infectious virions owing to the lack of Env. It has been consistently shown in other studies that HIV-2 progeny infectivity is much lower than HIV-1 when the virus produced from MDMs and HIV-2 infection of MDM can produce poorly infectious progeny virions (Refs. 37, 38). These previous findings are consistent with those from our study. Although HIV-2 could replicate in macrophages, endogenous LPTM5 targeting Env may result in the poorly infectious HIV-2 progeny particles. Thus, macrophages produce poorly infectious progeny virions during HIV-2 infection. Additional studies are required to investigate the effect of endogenous LPTM5 on HIV-2 Env in macrophages in the future.

Reviewer #6 (Remarks to the Author):

Comment 1: Reviewer 3 questions whether the localization of Vpr (primarily nuclear) makes sense given its effects on LPTM5, which is cytoplasmic. The authors respond with a literature review and extended data Figure 5. While the authors succeed in demonstrating that the localization of Vpr is complex and not well-defined, the overarching concern is not well-addressed. In addition to Vpr being nuclear, its cellular cofactors (E3-ubiquitin ligase CUL4A-DDB1 DCAF) and other reported targets have all been nuclear. Thus, the concern remains.

Response 1: Thank you for this interesting question. We are happy to address it.

First, from the beginning we did not exclude the possibility that Vpr and DCAF1 could be localized in the nucleus. Vpr is a shuttling protein (Ref. 7, PMID: 21831263), and we also noticed the presence of Vpr in the proximity of the nuclear membrane (Ext Fig. 4e, this revision). Therefore, we never concluded that Vpr could not be localized in the nucleus in macrophages.

Next, what we convey in this study is that cytoplasmic Vpr could be localized with LPTM5 in macrophages under the experimental conditions of our study. As for DCAF1, it also appears to be localized both in the cytoplasm and nucleus (PMID: 18524771, see Fig. 5A and B in their paper). In principle, because all proteins are synthesized in the cytoplasm, it is not surprising that even nuclear proteins show some cytoplasmic patterns. As an example, activation-induced cytidine deaminase (AID) is bona fide a nuclear-localized protein that initiates immunoglobulin switch and somatic hypermutation of chromosomes in the nucleus (PMID: 23664375; PMID: 26898111); however, overall AID shows a predominant pattern in the cytoplasm (PMID: 14769937, see Fig.1 in their paper). In contrast, only nuclear AID is capable

of accessing the immunoglobulin genes in the nucleus, and AID thus must be a nuclear-localized protein. Additionally, it has been reported in recent studies that cytoplasmic AID protein could also have several biological activities that only occur in the cytoplasm in the cell, such as AID against HBV reverse-transcription in nucleocapsids (PMID: 25836330, PMID: 24025329; PMID: 23341589; PMID: 28779685), LINE1 retro-transposition (PMID: 19188259; PMID: 23133680), and KHSV replication (PMID: 24244169), among others. Thus, AID, a shuttling protein from the cytoplasm to the nucleus, shows both cytoplasmic and nuclear biological functions in the cell. Likely, HIV-1 host restriction factor SAMHD1 is also a nuclear-localized protein (PMID: 2269137, see Fig.1b; Fig.2b in their paper) functioning to suppress HIV-1 reverse-transcription in the cytoplasm in macrophages and DCs (PMID: 21613998, PMID: 21720370).

Therefore, it is not surprising that DCAF1 or the identified Vpr targets can localize both in the cytoplasm and nucleus and that both proteins show activity both in the cytoplasm and nucleus.

Comment 2: Reviewer 3 expresses concern about the surprising result that the replication curves for two different viruses match so closely point by point in figure 1D and extended figure 1. We agree that the curves plus knockdown are nearly perfectly overlaid, which does not seem likely based on experience with primary macrophages, which are famously heterogeneous in their responses. Moreover, Donors 3, 12 and 13 are shown but what about donors 1, 2, 4,5,7,8,9,etc.? This pattern is concerning. Was only a subset of selected data shown? The response of the authors was that they provided additional data for donors 12-15 but extended data Figure 1 only shows donors 6, 14, 15 and 16. These additional examples share the same appearance where the curves with knockdown are nearly perfectly overlaid. The additionally data provided for the reviewers showing that a 50% knockdown has little to no significant effect doesn't reassure given the apparent selectivity with which the authors are revealing information.

Response 2: Thank you for this question.

We did not select any donors in this study. In our last submitted manuscript, we showed the previous data available for donors-3,4,5. We then followed the suggestion of the previous reviewers to perform new experiments associated with macrophages generated from other independent donors and subsequently replaced the previously obtained data for donors-4,5 and 7,8,9 with these new data. Thus, the numbers randomly assigned to donors-1,2,3,4,5,6...and so on simply mean that the results obtained for the macrophages derived from monocytes were isolated from respective independent healthy donors. Thus, they do not mean that we specifically selected any of them.

Comment 3: Reviewer 3 is also concerned about how the results in this paper fit with the literature regarding other restriction factors reported to be reversed by Vpr. The authors responded by suggesting that LPTM5 could be a central molecule in the previously reported mannose receptor-mediated restriction of HIV Env but they provide no evidence to support this assertion.

Response 3: Thank you for these comments. We are happy to give a response.

Because nuclear Vpr is proposed (Ref. 39) to inhibit mannose receptor (MR) protein transcription, thus reducing its cytoplasmic protein levels, together with their findings, we assumed that LPTM5 might be the central molecule involved in the MR pathway. Our assumption is only expressed in the response to the comments from reviewer-3 comments, and we did not state this hypothesis in the manuscript. The MR pathway in the cell is much more complicated, given that MR protein also counteracted by Nef protein. As for your suggestion, we could rephrase our statement to say that LPTM5 is an “important molecule” (rather than a “central molecule”), with the assumption that LPTM5 might be an important molecule involved in the MR pathway. Further research is needed to confirm this assumption.

Comment 4: Reviewer 3 is concerned about prior reports that LPTM5 is expressed in CD4+ T cells. The author’s response, that T cell activation downmodulates LPTM5 appears to be sufficient to alleviate the concern.

Response 4: Thank you for these comments.

Comment 5: Reviewer 3 is concerned that HIV-2 Vpr does not degrade LPTM5 as HIV-2 would also need to get around the same restriction. The authors speculate that HIV-2 may have another way to get around this restriction. While this is not very satisfying, it is conceivable and probably beyond the scope of this paper.

Response 5: Thank you for the helpful comments. We have addressed this issue in Response-3 to Reviewer-5.

Additional Comments– Many controls are missing and some data are concerning:

1. Figure 1 and extended Figure 1 are problematic with respect to the conclusion that decline in LPTM5 is Vpr-dependent. In the MDM system, the proportion of cells infected by WT virus is much greater than that by the Vpr-mutant. There is no control for these drastic differences in proportion of infected cells. This would be made obvious by including Gag on the western blot or including flow cytometric assessment of the proportion of cells that were infected based on

intracellular Gag stain. Thus, the alternative hypothesis that Vpr affects LAPTM5 indirectly by increasing the infection rate is not excluded.

Response 6: Thank you for the helpful comments.

In our Response 3 to Reviewer 2 in a previous version of the manuscript, we had stated that the presence of Vpr could promote the spread of HIV-1 infection, resulting in higher levels of infected cells. In fact, we know it is true.

First, as stated in the beginning of the manuscript (see Fig.1a), we used a lentiviral vector of FLAG-tagged Vpr to transduce macrophages in the absence of HIV-1, and this system has no connection to HIV infection. We have shown that LAPTM5 interacts with Vpr protein only in the presence and not in the absence of MG132. (Please also refer to the data in Supplementary Table 1, which were derived from two independent studies of a total of 10 healthy donors). These results suggest that Vpr could directly be associated with LAPTM5 protein in macrophages.

Second, our conclusion was not based on only one experiment. The presence of Vpr promotes LAPTM5 degradation (Fig. 1b), and Vpr protein associates with LAPTM5 only in presence of MG132 (Fig.1a). According to the data presented in Fig.2d, Fig.3b, Ext. Fig.3, 4, among others, it is clear that we have considered all the data and concluded that Vpr promotes degradation of LAPTM5. Moreover, even if we were only to think of the data in Fig. 1b, the proposal that the presence of Vpr promotes LAPTM5 degradation is reasonable. For example, we know that Vpr+ HIV-1 has higher infection ratio of MDM than does Vpr- HIV-1. Even if a high level of HIV-1 infection results in LAPTM5 degradation in MDM, our proposal is also reasonable, given that a higher level of HIV-1 infection is caused by the presence of Vpr. The only remaining question (only considering Fig.1b) is whether Vpr indirectly or directly promotes LAPTM5 degradation. However, we did not draw our conclusions solely based on the data shown in Fig.1b. As can be seen in Fig.1a,b; Fig.2c,d, Fig.3b,c; Ext Fig.3,4,5, among others, it is clear that Vpr directly promotes LAPTM5 degradation, which, therefore, does not occur owing to higher levels of HIV-1 infection of MDMs.

Third, as shown in Fig. 6b,c, we have used raltegravir to restrict Vpr+/- HIV-1 to a single-cycle infection of MDM, and thus Vpr+/- HIV-1 could result in similar Gag+ MDM cells (with a similar infection ratio) owing to the absence of a spreading infection. The HIV-1 Vpr+/- infection resulted in different levels of Env, whereas their infection ratio (as assessed by p24 ELISA) were similar in the single-cycle infection of macrophages. This would indicate that our conclusion is reasonable.

Lastly, in our previous responses to reviewers, we had described how we had sorted the same number of Gag⁺ MDM cells spreadingly infected with Vpr^{+/-} HIV-1 and LAPTMS5 is also degraded in presence of Vpr⁺ HIV-1 in contrast to Vpr⁻ HIV-1 with identical levels of Gag. We explain this further in the following figure only for reviewers:

(Figure caption: MDM infected with replication-competent 500 ng wild-type or Vpr-defective HIV-1 for 8 days. The p24-positive cells were sorted between Vpr^{+/-} HIV-1 infected MDMs. The same number of cells were lysed for western blotting to assess LAPTMS5, Gag, and GAPDH by their specific antibodies.)

Apparently, the same Gag⁺ positive cells infected with wild-type or Vpr-defective HIV-1 showed different levels of LAPTMS5, and the presence of Vpr did indeed promote LAPTMS5 degradation in macrophages.

Moreover, we also set similar Gag protein levels between Vpr⁺ and Vpr⁻ HIV in western blot experiments (the infected cells were not sorted) as shown in the following figure.

(Figure caption: MDM infected with replication-competent 100 ng wild-type or Vpr-defective HIV-1. Cell lysates from wild-type HIV-1 were diluted as indicated for western blotting to assess LAPTMS5, Gag, and GAPDH by their specific antibodies.)

As indicated by the arrow, LAPTMS5 protein was reduced by wild-type HIV-1 in contrast to Vpr-defective HIV with similar levels of infection (similar Gag protein levels). Taken together, the results indicate that Vpr could directly promote LAPTMS5 degradation.

2. Figures 2 and 3 as well as their extended figures show overexpression studies using HeLa cells in which they claim LAPTM5 decreases virion infectivity. The results show decreased viral infection following treatment of an indicator line with HeLa cell supernatant plus and minus HeLa cell transfection with LAPTM5. However the controls are insufficient to conclude that LAPTM5 decreases virion infectivity. Based on the data shown, the results could either be due to decreased virion release or a decrease in per particle infectivity. This could be resolved by including the p24 ELISA/Env ELISA shown in later figures. Figure 4 is better, but still lacks Env data.

Response 7: Thank you for the comments. Please allow us to address these issues.

First, as we stated with reference to Fig.2b, we used NL4-3-GFP combined with different Env. The fact that the NL4-3-GFP with VSV-G was not affected by LAPTM5 suggests that LAPTM5 did not affect HIV-1 release. This is considering that, LAPTM5 should affect the rate of infectivity of all NL4-3-GFP virions combined with different Env of HIV isolates except VSV-G.

In general, we have used p24 ELISA to show that LAPTM5 does not inhibit HIV release in producer cells. Importantly, we have shown that LAPTM5 did not affect virion release but inhibited virion infectivity (Fig.4). We have also concomitantly presented Gag and Env levels (Fig.5) to demonstrate that LAPTM5 reduces Env. Moreover, as shown in Figs.6 and 7, we concurrently measured levels of Env and Gag. When we considered these data together, we concluded that LAPTM5 inhibits virion infectivity by reducing Env.

3. Extended figure 2c shows LAPTM5 expression in THP1 cells. It is curious that no experiments were done in this cell line that naturally expresses LAPTM5.

Response 8: Thank you for these comments.

Indeed, we had presented data for THP-1 in the previously submitted manuscript. Following the reviewers' recommendation, we have deleted it in the current manuscript. The following figure does not appear in the manuscript but we are glad to include it here in response to your comment.

4. Figure 3 Part e has the same problem as for Figure 1 – however this figure is better because the authors are now showing Gag expression. This makes it clear that Gag expression is clearly lower in Vpr negative infected MDMs as you would expect given the expected lower proportion of infected cells in the Vpr-negative infection condition. Based on these results, one can't conclude reduction of LAPT5 is due to Vpr directly. It could be playing a role indirectly through increasing the proportion of infected cells in the sample.

Response 9: Thank you for the helpful comments. We believe that we have addressed this question in Response 6 to Reviewer 6.

5. Figure 5 Shows over expression of LAPT5 in HeLa cells leads to loss of Env but does not show this is reversible by Vpr. Everything is in a Vpr-negative background. No evidence that LAPT5-dependent targeting of Env to lysosomes is reversed by Vpr in part g. This is really important to show.

Response 10: Thank you for these comments.

As for Fig. 5, all of data were derived from the overexpressed but not endogenous LAPT5 in HeLa cells (endogenous LAPT5 is not detected in HeLa), and we described in the text (line 232-235, page 8) “Regardless of the presence of Vpr, LAPT5 overexpression also showed a robust inhibitory effect on wild-type HIV-1 infectivity (Extended Data Fig. 6a),

suggesting that the effect of Vpr expressed from HIV-1 was masked by the high levels of LAPTM5 protein.” As shown in Ext data Fig.6 in this revision, overexpressed LAPTM5 could mask the “endogenous” Vpr present in HIV-1. Therefore, it is not necessary to use wild-type Vpr+ HIV-1 with overexpressed LAPTM5.

Moreover, we think it is more important and reasonable to examine whether physiological endogenous LAPTM5 inhibition is reversed by Vpr in HIV-1. As shown in Fig.6a-c, we showed these data that Vpr in HIV-1 could reverse the Env from endogenous LAPTM5 inhibition in macrophages.

6. Figure 6 infection rates are missing– what proportion of the single round infections were Gag+? This is important because Vpr-dependent restriction factors can be overcome by high multiplicity of infection. Does knockdown of LAPTM5 increase proportion of infected cells in single round infections shown in Figures 6b and C? Figure 6d should include data demonstrating that targeting of Env to lysosomes is reversible by Vpr. This is really important to show.

Response 11: Thank you for these comments. Please allow us to explain further.

First, as shown in Fig. 6, we have presented results related to the low or high doses of HIV-1 infection in macrophages. Importantly, as shown in Fig.6b,c, Vpr showed a clear phenotype in our infection experiments, and its phenotype is not affected by high-dose infection. Therefore, we excluded the possibility that HIV-1 infection overcomes Vpr-dependent factors in our experiments. In addition, we described 10-ng (low dose) or 100-ng (high dose) infection of MDMs (see the figure legend), which normally give positive cells ranging approximately 1%–8% in single-cycle infection (no spreading infection occurs).

Second, because we have measured virion p24 levels (Fig.6b,c), knockdown of LAPTM5 did not affect p24, and thus it should not affect proportion of HIV-1 infected cells. Moreover, knockdown of LAPTM5 (Fig.6d) did not affect Gag levels as shown by western blot, indicating knockdown of LAPTM5 did not affect proportion of HIV-1 infected cells. Moreover, as shown in Figs.4a–c and Figs.5a–g, even overexpressed LAPTM5 did not affect HIV-1 Gag as measured by p24 ELISA.

Finally, as for Fig.6d, the reason we did not use Vpr+ HIV-1 is because Vpr in HIV could promote endogenous LAPTM5 degradation in macrophages. Therefore, shRNA of LAPTM5 and Vpr in HIV-1 have similar functions in reducing LAPTM5 protein levels, and their functions are redundant. Especially, we also used lysosome inhibitors in this experiment. Overall, it is not easy for us to compare shCtrl and shLAPTM5 samples when Vpr+ HIV-1 was used. Moreover, as shown in Fig.6b and c, we have indeed demonstrated that Vpr could reverse Env by subjecting

macrophages to a Vpr+/- HIV-1 infection. Therefore, we decided it was better to use a Vpr- HIV-1 infection with macrophages in this shRNA knockdown experiment (as shown in Fig. 6d).

7. Figure 7 Controls and obvious experimental results are missing – Need to round out the story by demonstrating that LAPTM5 targets Env for degradation and that this is reversible by Vpr for BOTH cell types. Key missing controls includes the proportion of infected cells for each experiment, which can easily be done with intracellular Gag staining.

Response 12: We are happy to address your comment.

First, we have indeed demonstrated that Vpr in HIV-1 reversed Env levels but not Gag levels (Fig.7b and c), as we measured p24 (for Gag levels) and gp120 (for Env levels) ELISA.

Second, virion p24 levels could reflect a proportion of HIV-infected cells, thereby suggesting LAPTM5 did not influence Gag+ cells (TZM-bl) (as shown in Fig.7b and c). With respect to the CD4+ T cells infected with replication-competent 89.6, p24 ELISA also represents intracellular Gag levels. Therefore, we think is unnecessary to stain intracellular Gag for both cells.

In addition, as shown in Fig.5e and f, we have concomitantly stained the intracellular Gag and Env as well as LAPTM5. Moreover, we have also presented Gag protein levels in western blots or p24 ELISA (Fig.5a–g, and Fig.6b–d.), which is similar to intracellular Gag staining. We hope that our explanation could be acceptable by reviewer-6.

8. I am concerned about the western blots shown in Figure 7a and 7d (full blots are shown in file named 260399_1_data_set_4992922_qhx6sf.pdf). It doesn't make sense that mock treated TZMBL and primary T cells, which are very different kinds of cells have the same cross-reacting lower background band and that in both cases this same background band mysteriously goes away in LAPTM5 transfected/transduced cells. Given the pattern shown, I am suspicious that macrophage lysate was substituted for the transduced/transfected cells and mislabeled in the figure.

Response 13: Thank you for these comments. Please allow us to explain further.

Apparently, the non-specific bands in the mock vector-treated TZM-bl or CD4+ T cells should somehow be attributed to the transfected mock plasmid treatment and not to the presence of different cells. Because the LAPTM5 antibody we used in this study is a polyclonal antibody derived from rabbit (and can react with human cells), these non-specific signals may occasionally be observed if the blots are not perfectly clean or if the skimmed milk or antibodies in PBST were reused repeatedly. Similarly, we can also see lower non-specific bands on GAPDH western blots. Perhaps, this is also due to the above reasons. To correct this, we have

performed a new western blot with fresh antibodies, fresh BSA in PBST. The resulting blots are cleaner, as shown in Fig.7a and d.

Reviewers' comments:

Reviewer #2 (Remarks to the Author):

I have been asked to comment on the issues raised by Reviewer #6.

The manuscript is full of beautiful data, but, indeed, some points could be improved. More specifically, regarding Reviewer #6 's comments:

1. I feel that localization concerns are beyond the scope of this manuscript.
2. Reviewer #6 believes there has been some sort of selectivity with which the reviewers are revealing information because not all macrophage donors have been presented. If this were the case, I believe donors would have been labelled differently. I feel a lot of donors have already been presented.

Nonetheless, I cannot explain why the curves are so well overlaid in Figure 1f (and extended) (curve shLAP-1 Ad8 Vpr (-) perfectly overlaid with curve Ad8 Vpr (+).)

3. To see whether the results in this paper fit with the literature regarding other restriction factors seem also beyond the scope of this study. Asking about the mannose receptor is a little hard as this refers to a manuscript published in 2020.

6. Reviewer #6 would like to see that targeting Env to lysosome is reversible by Vpr in Figure 6d. I do agree that this would be a real “plus”. Figure 6d could have been done with AD8 Vpr+. The authors explain they feel it is the same with shLAPTM5 that mimics Vpr, but this is not the same.

7. We have no idea of the proportion of infected cells in Figures 6 and 7, this kind of information would be of value to speak about infectivity.

Reviewer #6 (Remarks to the Author):

The authors have not provided any new data that changes my view of their submission. My concerns remain. As I previously wrote many of the experiments have problems with rigor that they authors have tried to rationalize but have not adequately addressed. For example, demonstrating that GFP is equal by western blot is not the same as showing that equal p24 has been released from the cells and adjusting supernatants such that equal p24 is added to an infectivity assay so that you can conclude that infectivity rather than viral particle production is indeed affected. This is very basic for virological studies. Results from all donors should be shown either as primary data or compiled. There is too much selected presentation of results and scattered application of standard controls for me to feel comfortable with this manuscript.

Response to Reviewers' Comments

We thank the reviewers for their helpful comments and suggestions, which have helped us significantly improve the manuscript. All changes are marked in red in the revised manuscript, and our point-by-point responses to the reviewers' comments are presented below.

Reviewers' comments:

Reviewer #2 (Remarks to the Author):

I have been asked to comment on the issues raised by Reviewer #6.

The manuscript is full of beautiful data, but, indeed, some points could be improved. More specifically, regarding Reviewer #6 's comments:

1. I feel that localization concerns are beyond the scope of this manuscript.

Response 1: Thank you for your comments.

2. Reviewer #6 believes there has been some sort of selectivity with which the reviewers are revealing information because not all macrophage donors have been presented. If this were the case, I believe donors would have been labelled differently. I feel a lot of donors have already been presented.

Nonetheless, I cannot explain why the curves are so well overlaid in Figure 1f (and extended) (curve shLAP-1 Ad8 Vpr (-) perfectly overlaid with curve Ad8 Vpr (+).)

Response 2: Thank you for your comments. As for the donors, we did not have any overlapping data between them. The Ext data Figs. 1f-g in the previously submitted paper was included in the new Figs.1b-c in this revision. We have renumbered all donors and shown them all in a compiled primary figure in Fig. 1 of this revision. The knockdown results are included in Ext data Fig.1e-f in this revision. Moreover, the current donor-1,2,3,4,5,6, and 7 in this revision are previously numbered donors -3,12,13,6,14,15,and 16, respectively in the previously submitted manuscript. The donor-number 8, in Fig.1d-e is labeled as in the previously submitted manuscript of Figs.1e-f we did not indicate the donor's number.

3. To see whether the results in this paper fit with the literature regarding other restriction factors seem also beyond the scope of this study. Asking about the mannose receptor is a little hard as this refers to a manuscript published in 2020.

Response 3: Thank you for your comments.

6. Reviewer #6 would like to see that targeting Env to lysosome is reversible by Vpr in Figure 6d. I do agree that this would be a real "plus". Figure 6d could have been done with AD8 Vpr+. The authors explain they feel it is the same with shLAPTM5 that mimics Vpr, but this is not the same.

Response 4: Thank you for your comments. As for Vpr reversing the loss of Env and infection ratios in Figs.6, and 7, we have followed your suggestions and performed new experiments

utilizing both Vpr⁺ and Vpr⁻ HIV-1 to infect macrophages and to elucidate whether Vpr's presence could reverse the LAPTM5-mediated loss of Env. As a result, we have shown that Vpr does indeed reverse the loss of Env LAPTM5 targets in macrophages, as demonstrated in Fig. 6d and Figs. 7b-c in this revision. Moreover, we have also addressed this data in the main text of the manuscript as follows '*Initially, the levels of Gag protein in the cell lysates were similar between wild-type and Vpr-defective HIV-1 (Fig. 6d), indicating similar infection ratios of MDMs.*' (Page 10, Lines 291-293), and '*Moreover, similar levels of Gag protein in cell lysates indicated a comparable infection ratio between wild-type and Vpr-defective HIV-1 in this single-cycle infection.*' (Page 11, Lines 325-327). Importantly, as shown in Fig. 6d and Figs. 7b-c, the presence of Vpr in wild-type HIV-1 can reverse the LAPTM5-mediated reduction of Env (lanes 1 vs 5 in Fig.6d; lanes 1 vs 2 or lane 5 vs 6 in Figs.7b-c) in comparison to Vpr-defective HIV-1. Moreover, LAPTM5 protein is consistently downregulated in the presence of Vpr (lanes 1 or 3 vs 5 or 7 in Fig.6d; lanes 1 vs 2 or lanes 5 vs 6 in Figs.7b-c). We have now re-written these paragraphs to present the data more cohesively (Pages 10-11, Lines 287-337).

7. We have no idea of the proportion of infected cells in Figures 6 and 7, this kind of information would be of value to speak about infectivity.

Response 5: Thank you for your comments. First, we have followed your suggestion and performed new experiments under the same condition for Figs. 6 and 7 to measure the proportion of infected macrophages, and demonstrated a similar infection ratio between Vpr⁺ and Vpr⁻ HIV-1 (Ext Figs. 9a,b) in this revision.

Next, we also measured the Gag protein levels in cell lysate both for Fig. 6d and Figs.7b,c. This is to demonstrated that the wild-type and Vpr-defective HIV have similar infection ratios both in macrophages and TMZ-bl cells in single-cycle infections. We have also described this data in the main text of the manuscript as follows '*Initially, the levels of Gag protein in the cell lysates were similar between wild-type and Vpr-defective HIV-1 (Fig. 6d), indicating similar infection ratios of MDMs.*' (Page 10, Lines 291-293), and '*Moreover, similar levels of Gag protein in cell lysates indicated a comparable infection ratio between wild-type and Vpr-defective HIV-1 in this single-cycle infection.*' (Page 11, Lines 325-327) in this revision.

Reviewer #6 (Remarks to the Author):

The authors have not provided any new data that changes my view of their submission. My concerns remain. As I previously wrote many of the experiments have problems with rigor that they authors have tried to rationalize but have not adequately addressed. For example, demonstrating that GFP is equal by western blot is not the same as showing that equal p24 has been released from the cells and adjusting supernatants such that equal p24 is added to an infectivity assay so that you can conclude that infectivity rather than viral particle

production is indeed affected. This is very basic for virological studies. Results from all donors should be shown either as primary data or compiled. There is too much selected presentation of results and scattered application of standard controls for me to feel comfortable with this manuscript.

Response 1: Thank you for your constructive comments to improve our manuscript. We have incorporated your suggestions and have included all data you asked for in this revision, and we have addressed this in further detail below.

(1). Where Fig. 4 was missing the Env data in the previously submitted manuscript, we have now included gp120 WB data to substantiate LAPTM5 against HIV-1 infectivity via the reduction of Env as shown in Fig. 4 (Western blotting) in this revision. As a result, LAPTM5 reduced Env in cell lysates, indicating that the inhibition of infectivity is due to the loss of Env.

(2). To demonstrate that Vpr reverses the loss of Env in Fig. 6d of the previously submitted manuscript, we have used both Vpr+ and Vpr- HIV-1 to infect macrophages. At first, we have assessed Gag protein levels to demonstrate similar infection efficiencies between wild-type and Vpr-defective HIV-1 in a single-cycle infection. Accordingly we have also edited our explanation in the main text in this revision to be following *‘Initially, the levels of Gag protein in the cell lysates were similar between wild-type and Vpr-defective HIV-1 (Fig. 6d), indicating similar infection ratios of MDMs.’* (Page 10 Lines 287-315). Next, as shown in Fig. 6d in this revision, the presence of Vpr did indeed reverse the loss of Env LAPTM5 targets in macrophages, in contrast to Vpr-defective HIV-1 (lanes 1 vs 5 in Fig.6d). Accordingly, LAPTM5 protein decreased in the presence of Vpr (lanes 1 or 3 vs 5 or 7). In addition, Env was not detected within lysosomes in the presence of LAPTM5 without lysosomal inhibition (lanes 9, 13); however, lysosomal inhibition dramatically increased Env in lysosomal lysates in the presence of LAPTM5 (lanes 11, 15). We have re-written these paragraphs to clearly explain these results (Pages 10-11, Lines 287-315). Taken together, the presence of Vpr reverses the loss of Env that LAPTM5 targets during HIV-1 infection of macrophages.

(3). To conclusively demonstrate that Vpr reverses Env in the previous Figs. 7b-c. we have performed Western blotting to measure gp120, Gag, LAPTM5 and GAPDH expression levels in TZMbl cells infected with wild-type and Vpr-defective HIV-1. This data is shown in Fig. 7b-c (Lanes 1 vs 2 or Lane 5 vs 6) in this revision, and we have concluded in the main text of the manuscript as follows: *‘Moreover, similar levels of Gag protein in cell lysates indicated a comparable infection ratio between wild-type and Vpr-defective HIV-1 in this single-cycle infection.’* (Page 11, Lines 325-327). As shown in Figs. 7b-c in this revision, the presence of Vpr in wild-type HIV-1 reverses the LAPTM5-mediated reduction of Env (lanes 1 vs 2 or lane 5 vs 6) in comparison to Vpr-defective HIV-1. In addition, gp120 ELISA derived from culture supernatants also consistently revealed that the presence of Vpr reverses Env in HIV virions. Moreover, LAPTM5 protein is also downregulated in the presence of Vpr (lanes 1 vs 2 or lanes 5

vs 6 in Fig.7b,c). As stated above, we have now reworded the paragraph to present the new data (Page 11, Lines 318-337). In summary, the presence of Vpr could reverse the loss of Env during HIV-1 infection of TZMbl cells.

(4) Next, in Fig. 7e, as this is an HIV-1 spreading infection system in CD4⁺ T cells, p24 ELISA results also revealed some different Gag levels after a 6-day culture in the absence of LAPTM5. This may be due to Vpr inducing dividing cells to undergo cycle arrest. We have followed your suggestion and measured the proportion of infected cells on days 2, 4 and 6 now included in Ext Fig.9b in this revision. Although slightly different infection ratios can be seen in 6 day cultures in the absence of LAPTM5, the dramatic differences of p24 ELISA between Vpr⁺ and Vpr⁻ HIV-1 in the presence of LAPTM5 are not due to these different infection ratios between Vpr⁺ and Vpr⁻ HIV-1. Moreover, in the 4 day infected period, the infection ratio between Vpr⁺ and Vpr⁻ HIV-1 are almost similar in the absence of LAPTM5. In contrast, the presence of LAPM5 dramatically inhibited Vpr-HIV-1 than Vpr⁺ HIV⁻.

Moreover, we have also measured the infected cell population in the single-cycle infection system as shown in Ext Fig.9a that also shows almost the same infection ratios between Vpr⁺ and Vpr⁻ HIV-1. Taken together, our conclusion was made by using the infection system with a similar infection ratio between wild-type and Vpr-defective HIV-1.

(3). We agree with this reviewer that the overlay in donors may raise a concern, but it is *de facto* not the case in our study. All of the different donors simply mean that they are independent experiments. For example, in our previous paper (PMID: 31061530), we have used a total of 57 donors' CD4⁺ T cells for experiments and the number of donors is not a consecutive number (such as from 1, 2, 3, 4, 5, 6..... to 57). During revising this manuscript, if the data was deleted for donor-2 and replaced with new data derived from another new healthy donor (such as donor-37), we have to insert the data of donor-37 to where data of donor-2 was. Then, the donor number will be: donor-1, donor-37, donor-3. This is the same explanation for our previously submitted manuscript. This may cause confusion and as per your request, we have renumbered all of the donors and updated the data in the compiled primary figure (Fig. 1) in this revision. The Ext data Figs.1f-g in the previously submitted paper are now included in the new Fig.1b-c in this revision. The knockdown results were included in Ext data Fig. 1e-f in this revision. Moreover, the current donor numbers-1,2,3,4,5,6, and 7 in this revision are the previous donor numbers 3,12,13,6,14,15, and 16, respectively in previously submitted manuscript. The donor-8 in Figs.1d-e is now labeled, and in previously submitted manuscript Fig.1e-f we did not indicate the donor's number.

REVIEWERS' COMMENTS

Reviewer #2 (Remarks to the Author):

I believe the authors have correctly answered the previous comments. Of importance, they now present new data showing that LPTM5-mediated Env targeting to lysosomes is reversible by Vpr expressed from macrophage-tropic HIV-1. Consistently, in these experiments, LPTM5 is downregulated in the presence of Vpr. Overall, the manuscript is very interesting and the data are very impressive.